# AMPA receptor GluA2 subunit defects are a cause of neurodevelopmental disorders

Vincenzo Salpietro et al.[#]

AMPA receptors (AMPARs) are tetrameric ligand-gated channels made up of combinations of GluA1-4 subunits encoded by *GRIA1-4* genes. GluA2 has an especially important role because, following post-transcriptional editing at the Q607 site, it renders heteromultimeric AMPARs $Ca^{2+}$-impermeable, with a linear relationship between current and trans-membrane voltage. Here, we report heterozygous *de novo GRIA2* mutations in 28 unrelated patients with intellectual disability (ID) and neurodevelopmental abnormalities including autism spectrum disorder (ASD), Rett syndrome-like features, and seizures or developmental epileptic encephalopathy (DEE). In functional expression studies, mutations lead to a decrease in agonist-evoked current mediated by mutant subunits compared to wild-type channels. When GluA2 subunits are co-expressed with GluA1, most *GRIA2* mutations cause a decreased current amplitude and some also affect voltage rectification. Our results show that *de-novo* variants in *GRIA2* can cause neurodevelopmental disorders, complementing evidence that other genetic causes of ID, ASD and DEE also disrupt glutamatergic synaptic transmission.

Abnormal glutamatergic synaptic transmission and plasticity has been implicated in some neurodevelopmental disorders (NDDs) featuring intellectual disability (ID), developmental delay (DD), and autism spectrum disorders (ASDs)[1–5], as exemplified by the identification of rare de-novo mutations in genes encoding ionotropic glutamate receptor (iGluR) subunit genes[6–13]. iGluRs are the major mediators of fast excitatory neurotransmission in the vertebrate brain[13–17]. They include N-methyl-D-aspartate receptors (NMDARs), kainic acid receptors (KARs), and α-amino-3-hydroxy-5-methyl-4-isoxazole propionic acid receptors (AMPARs)[14,18]. AMPARs are assembled from four subunits (GluA1-4), with GluA1/GluA2 heterotetramers being the most frequent combination in the forebrain[19]. Mutations in GRIA1, GRIA3, and GRIA4 (encoding GluA1, GluA3, and GluA4) have been established as very rare causes of NDDs[9,10,20,21]. The GluA2 subunit, encoded by GRIA2, has a major role in the regulation of AMPAR $Ca^{2+}$ permeation and voltage rectification, in large part mediated by an arginine residue in the ion-selectivity filter that results from post-transcriptional editing of a codon for glutamine[22,23]. Hitherto, evidence for a role of GluA2 in NDDs has been sparse. A microdeletion case report previously suggested a possible link between GRIA2 haploinsufficiency and ID[24], and an in-frame deletion of three amino acids was identified in one individual recruited in the Deciphering Developmental Disorders (DDD) study (https://www.ddduk.org/), containing exome sequencing data from over 13,000 individuals affected with developmental disorders[3]. Furthermore, abnormal translation of the AMPAR GluA2 subunit via changes in GRIA2 expression or alternative splicing has been implicated in the pathophysiology and the neurological phenotype of a wide array of NDDs, including Fragile X syndrome (FXS) and Rett syndrome (RTT)[25–27]. Further underlining the potential importance of GluA2 for normal CNS development and function, $Gria2^{-/-}$ mice have increased mortality, show enhanced NMDAR-independent long-term potentiation, consistent with abnormal $Ca^{2+}$ permeation through GluA2-lacking AMPARs, and exhibit impaired motor coordination and behavioral abnormalities[28]. Despite the hints from the above studies, GRIA2 mutations have hitherto not been considered an important cause of human disease, and there is no established disease association for this gene in the Online Mendelian Inheritance in Man (OMIM) database (MIM #138247).

We report 28 unrelated individuals, affected with neurodevelopmental abnormalities encompassing ID/DD, ASD, RTT-like features and seizures or developmental epileptic encephalopathy (DEE), in whom we have identified heterozygous de novo variants in GRIA2. Functional analyses reveal loss of function for the majority of the mutations, supporting GluA2 defects as a cause of NDDs with variable associated neurological phenotypes.

## Results

**Identification of GRIA2 de-novo variants**. The index case was a 4-year-old boy diagnosed with DEE (Patient 1, Supplementary Table 1) who was found to carry a de-novo variant in GRIA2 by trio whole exome sequencing (WES). We next screened exomes and genomes (WGS) from the DDD Study and the SYNaPS Study Group (http://neurogenetics.co.uk/synaptopathies-synaps) and compared genetic datasets with collaborators and identified seven individuals carrying GRIA2 de-novo intragenic variants (Supplementary Table 1, Patients 2–8). Through further collaborations and research networks we ascertained sixteen additional individuals (Patients 9–25) carrying GRIA2 de-novo variants and three individuals (Patients 26–28) with de-novo 4q32.1 microdeletions leading to GRIA2 haploinsufficiency (Supplementary Fig. 1). In total, we found 20 different GRIA2 de-novo intragenic

variants including missense ($n = 15$), splice-site ($n = 2$), in-frame deletion ($n = 1$), stop-gain ($n = 1$) and frameshift ($n = 2$) variants (Supplementary Tables 1–2, Fig. 1a). Intragenic variants were first identified by WES, WGS or massively parallel targeted sequencing and confirmed as de-novo by trio Sanger sequencing in all patients (Methods, Supplementary Fig. 2). De-novo microdeletions were found by chromosomal microarray analysis (Patients 26–28) and validated using established laboratory protocols.

**GRIA2 is constrained and intolerant to loss-of-function**. In the Exome Aggregation Consortium (ExAC) database (http://exac.broadinstitute.org) GRIA2 is highly constrained for missense variation (z-score: 4.43) and intolerant to loss-of-function (LoF, intolerance score: 1.00)[29]. Among the affected probands reported here, we found several frameshift, stop-gain and splice-site variants and 4q32.1 microdeletions predicted to lead to GRIA2 haploinsufficiency (Supplementary Table 2). All the identified intragenic de novo variants were absent from Genome Aggregation Database (GnomAD, http://gnomad.broadinstitute.org) and ExAC and displayed high conservation (Fig. 1c, Supplementary Fig. 3) with a mean: GERP[++] score 5.51 and in-silico pathogenic predictor scores (mean: CADD_Phred 28.924). In total, in our cohort de-novo GRIA2 variants with predicted LoF were found in 8 out 28 patients. De-novo frameshift deletions in Patients 10 and 12 lead to changes in the reading frame with the generation of a premature stop at codon 37 and 14 amino acids downstream, respectively. In Patient 19, a single-nucleotide substitution leads to a stop-gain variant (p.R323ter). The three de novo 4q32.1 microdeletions identified in patients 26–28 encompass chr4:157,343,163–158,271,008 bp (GRCh37/hg19) as the smallest overlapping deleted genomic region, and within this region GRIA2 is the gene most intolerant to LoF (Supplementary Tables 3–5). De-novo splice-site variants in Patients 8 and 11 are predicted to cause loss of donor splice sites at exons 1 and 11, respectively, according to in-silico Alamut predictions (Supplementary Figs. 4–5)[30]. Of the other 19 patients, 18 harbored de novo missense mutations, and one had a 9-bp deletion predicted to lead to loss of amino acids 528–530.

**Phenotypic spectrum associated with GRIA2 de novo variants**. Consistent with the role of GluA2 channels in synapse development and plasticity[19], phenotypic analysis of patients carrying de-novo GRIA2 variants demonstrated an NDD spectrum including ID/DD, developmental regression, ASD, speech impairment, RTT-like features, and seizures or DEE (Fig. 1b; Supplementary Videos 1–5). Supplementary Table 1 summarizes the core phenotypic features of all 28 Patients, aged between 3 months and 31 years. In all cases, onset of GRIA2-related NDD occurred in childhood. Several individuals had normal early developmental milestones and started to exhibit variable impairment of motor coordination, social interaction, and language abilities in infancy (Supplementary Table 1, Supplementary Notes 2, 3, 4, and 16). In some affected individuals, social or language regression was reported (Patients 3 and 4). Between 2 and 6 years of age several patients developed RTT-like features (Supplementary Table 1, Supplementary Table 6), including stereotyped hand movements (Supplementary Notes 2, 3, 4, and 7), screaming episodes (Supplementary Notes 2, 3, 6, and 14), gait abnormalities including ataxia and dyspraxia (Supplementary Notes 4 and 6), abnormal sleep rhythm (Supplementary Notes 4 and 14), and irregular breathing patterns with hyperventilation episodes (Supplementary Note 7). Progressive microcephaly was observed in 4 out of 28 individuals (Supplementary Notes 6, 13, 17, and 21) with a deceleration of head growth usually occurring during infancy (Supplementary Table 1). Several patients were diagnosed with

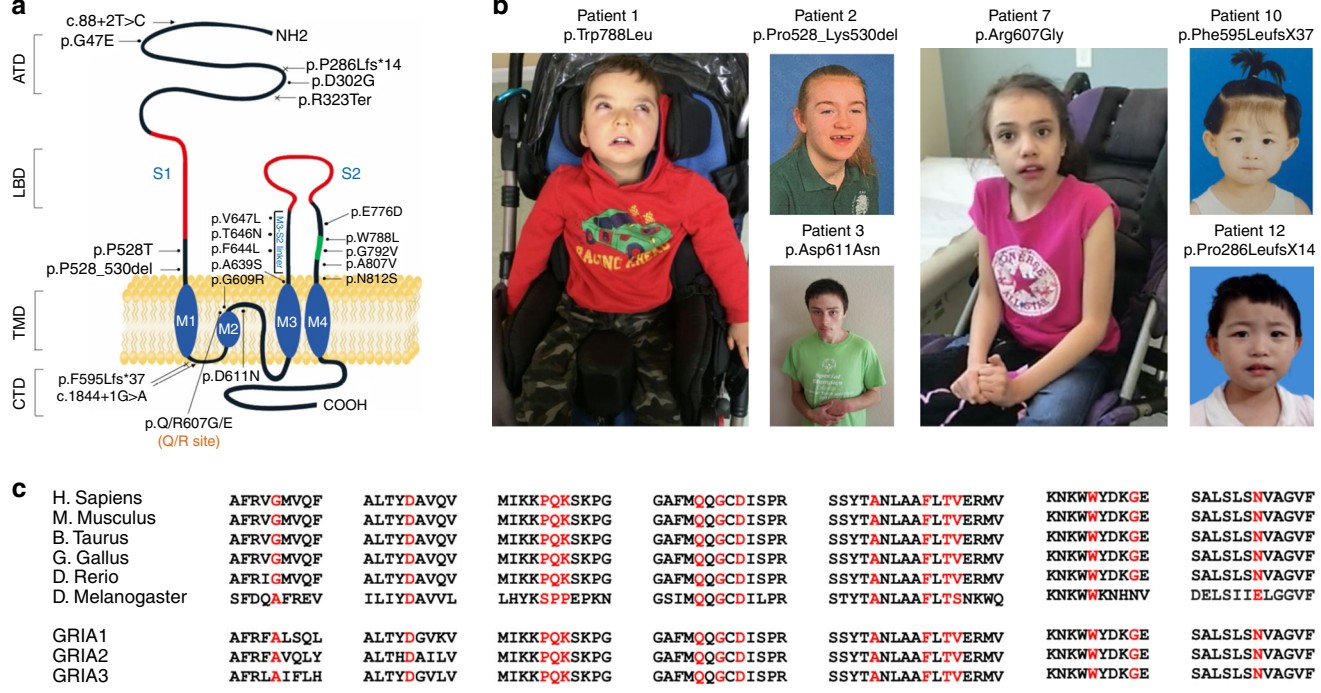

**Fig. 1** *GRIA2* intragenic de-novo variants identified in this study. **a** Schematic of the human GluA2 protein (NP_000817.2) indicating the positions of twelve missense changes (dot arrows), two frameshift deletions (cross arrows), two splice-site variants (arrows) and an in-frame deletion (dot arrow). Glutamate binding regions are displayed in red, flip/flop alternatively spliced region is represented in green. **b** Left panel: Patient 1, carrying the de-novo p.W788L GluA2$^{Flop}$ variant, at 3 years, exhibiting hypotonia and an oculogyric crisis; he is wheelchair dependent. Middle left panel: Patient 2 (top) carrying the de-novo p.P528_K530del in-frame deletion, at 12 years; Patient 3 (bottom) carrying the de-novo p.D611N variant, at 18 years, exhibiting hand-wringing suggestive of RTT. Middle right panel: Patient 7, carrying the de-novo p.Q607E/p.R607G heterozygous mutation (affecting the Q/R editing site) at 10 years, exhibiting hand-wringing as part of a RTT-like presentation. Right panel: Patient 10 (top) carrying the de-novo p.F595LfsX37 variant, at 5 years; Patient 12 (bottom) carrying the de-novo p.P286LfsX14 at 6 years. **c** Multiple alignment showing GluA2 protein complete conservation across species and inter AMPAR homolog subunits (GluA1, GluA3, and GluA4) alignment. Human GRIA2 (NP_000817.2), mouse GRIA2 (NP_001077275.1), bos taurus GRIA2 (NP_001069789.2), gallus gallus GRIA2 (NP_001001775.2), danio rerio (NP_571970.2), drosophila melanogaster (NP_476855.1), Human GRIA1 (NP_000818.2) Human GRIA3 (NP_015564.4), and Human GRIA4 (NP_000820.3)

ASD (Supplementary Notes 2, 3, 9, 10, 11, 12, 15, 19, 22, 23, and 26), and some presented repetitive behavior patterns and impaired social interaction (Supplementary Notes 8, 27, and 28). Language impairment was present in all patients, with the majority attaining no meaningful speech (Supplementary Table 1). Twelve patients suffered from seizures (Supplementary Notes 4 and 5) or DEE (Supplementary Notes 1, 7, 13, 16, 17, 18, 20, 21, 24, and 25) usually starting within the first 6 months of life, including infantile spasms, tonic-clonic, myoclonic and focal seizures (Supplementary Table 7). EEG features included polyspikes, slow spike and wave, and bilateral temporal non-synchronized epileptic activity. The clinical outcome was also variable (Supplementary Table 1, Supplementary Notes 1, 7, 13, 16, 17, 18, 20, and 24). MRI scans in 7 DEE patients showed progressive brain (mainly cerebellar) atrophy and white matter abnormalities in some (Fig. 2; Supplementary Table 1).

**AMPAR Molecular dynamic stimulations**. To compare the structural mobility of GluA2 and its mutants we built a model of each protein ectodomain including a ligand-binding domain (LBD) and an amino-terminal domain (ATD; Methods) by modeling mutations on the wild type and followed their behavior along time by means of atomistic molecular dynamics simulations in water solvent. Proteins with mutations in the pore region where omitted as either the mutations were close to or included in the transmembrane domains (TMDs) which were not modeled. We aimed to ascertain the effect of the mutations on the glutamate (GLU) binding-pocket. Interestingly, in the studied mutated

proteins this group of atoms appear to have a higher level of rigidity compared to the wild-type protein (Figs. 3 and 4). Although the wild-type crystal structure is symmetric, after 10 ns the conformations of pockets associated with chains C and D, which are coupled in the binding site, diverge with respect to those associated with A and B which do not change conformation. In the observed timeframe molecules underwent concerted macroscopic movements and this is reflected by minor variations in their backbone root mean squared deviation (RMSD, Supplementary Fig. 9) and radius of gyration (Supplementary Fig. 10). The RMSD, which is a measure of the average atoms displacement from the starting configuration, clearly indicates that amino acids in the GLU binding site are independently mobile at a timescale consistent with our simulations with two pockets reaching values larger than 0.27 nm. However, the same is not true for most mutants: p.D302G, p.F644L, p.P528T, and p.V647L whose RMSD do not exceed 0.22 nm. The remaining mutants studied showed an intermediate behavior.

**Functional analyses of the identified *GRIA2* variants**. To assess the functional consequences of *GRIA2* missense mutations, we synthesized cDNA encoding the human GluA2 wild-type and mutant channels and transfected HEK293T cells together with the auxiliary stargazin protein (Methods). Except for one mutation (see below), amino acid position 607 was made to encode an arginine residue in GluA2 to mimic post-transcriptional editing of a genomic glutamine-encoding codon. Coding variants in the N-terminal domain, linkers (including the three amino acid

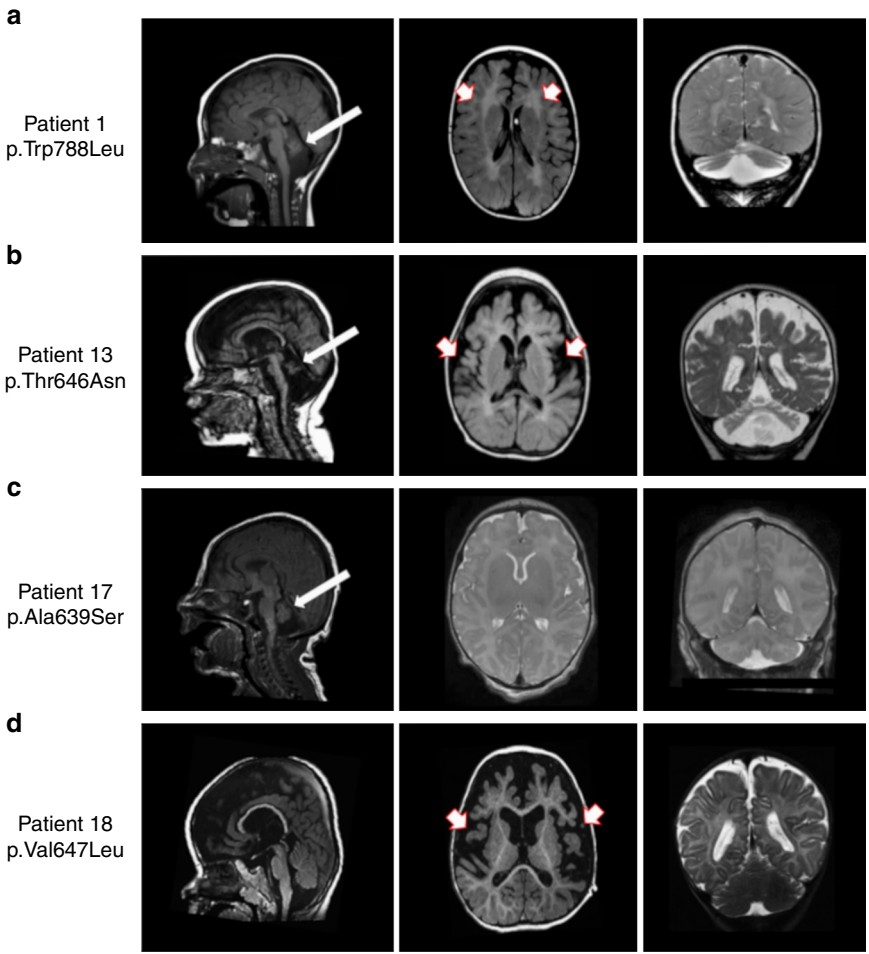

**Fig. 2** Brain imaging in 4 individuals with *GRIA2*-related DEE and brain and cerebellar atrophy. **a** Sagittal T1 weighted (left panel), Axial T1 weighted (middle panel), and **c**oronal T2 weighted (right panel) images from Patient 1 (carrying the de-novo p.W788L mutation) scanned at the age of 3 years. There is reduction in the cerebral white matter (red and white arrows), brain volume and marked cerebellar atrophy with vermian deficiency (white arrows). **b** Sagittal T1 weighted (left panel) images from Patient 13 (carrying the de-novo p.T646N mutation) at the age of 2 months, and axial T1 weighted (middle panel) and coronal T2 weighted images (right panel) from the same Patient at the age of 11 months demonstrate white matter signal abnormality (white and red arrows) with generalized reduction in the cerebral white matter volume and cerebellar atrophy with vermian deficiency (white arrows). **c** Sagittal T1 weighted (left panel), Axial T2 weighted (middle panel) and coronal T2 weighted (right panel) images from Patient 17 (carrying the de-novo p. A639S mutation) at the age of 6 days; there is an underdeveloped/hypoplastic cerebellum (white arrows) and delayed maturation of myelin. **d** Sagittal T1 weighted (left panel), axial T1 weighted (middle panel) and coronal T2 weighted (right poanel) images from Patient 18 (carrying the de-novo p.V647L mutation) scanned at the age of 18 months; there is a global cerebral atrophy and white matter changes which suggest hypomyelination (white and red arrows). Although the volume of the cerebellar hemispheres is preserved, atrophy of the inferior cerebellar vermis and wide cerebellar sulci are seen

deletion) and pore were selected for functional analysis. Inward currents evoked by the non-desensitizing agonist kainic acid (KA, 1 mM) on HEK cells expressing homomeric GluA2 were significantly decreased for 7 out of the 11 variants tested, including those associated with NDD or NDD and DEE (Fig. 5a). Three variants (p.P528T, p.D611N, and p.V647L) exhibited apparently normal current amplitudes. In contrast, the p.Q607E variant exhibited KA-evoked currents that were larger than the wild-type control. This mutation affects the Q607 codon that is normally edited to an arginine residue. The increase in current amplitude is consistent with removal of a positively charged residue from the ion conduction pathway.

GluA2 homomers are not thought to occur naturally. A common stoichiometry in the forebrain is channels composed of two GluA1 and two GluA2 subunits. Presence of the GluA2, edited at the Q607 site, in the heteromeric channel reduces the single channel conductance, confers $Ca^{2+}$-impermeability, and results in a linear current-voltage relationship[22,23]. In contrast, homomeric GluA1 channels exhibit a larger rectifying

conductance and are $Ca^{2+}$-permeable. We therefore repeated the functional studies co-expressing wild type or mutant GluA2 together with GluA1 and stargazin. Wild-type GluA2 co-expressed with GluA1 yielded approximately two-fold larger KA-evoked currents than GluA1 alone (Fig. 5b). Five of the mutants significantly decreased the KA-evoked current amplitude relative to wild-type GluA2. Interestingly, two of these variants (p. D302G and p.G609R) reduced the current below the level obtained with GluA1 alone, suggesting a dominant negative effect (Fig. 5b). Of the remaining variants, all but one (p.P528T) exhibited an apparent decrease relative to wild type, although falling short of significance.

We complemented the KA-evoked current amplitude measurements with assessment of rectification by ramping the holding voltage between −104 and +76 mV. We confirmed that GluA1 expressed alone yielded a doubly rectifying current-voltage relationship. It was linear when wild-type GluA2 was co-expressed (Fig. 6). Seven of the mutants significantly increased the degree of rectification compared to wild-type GluA2.

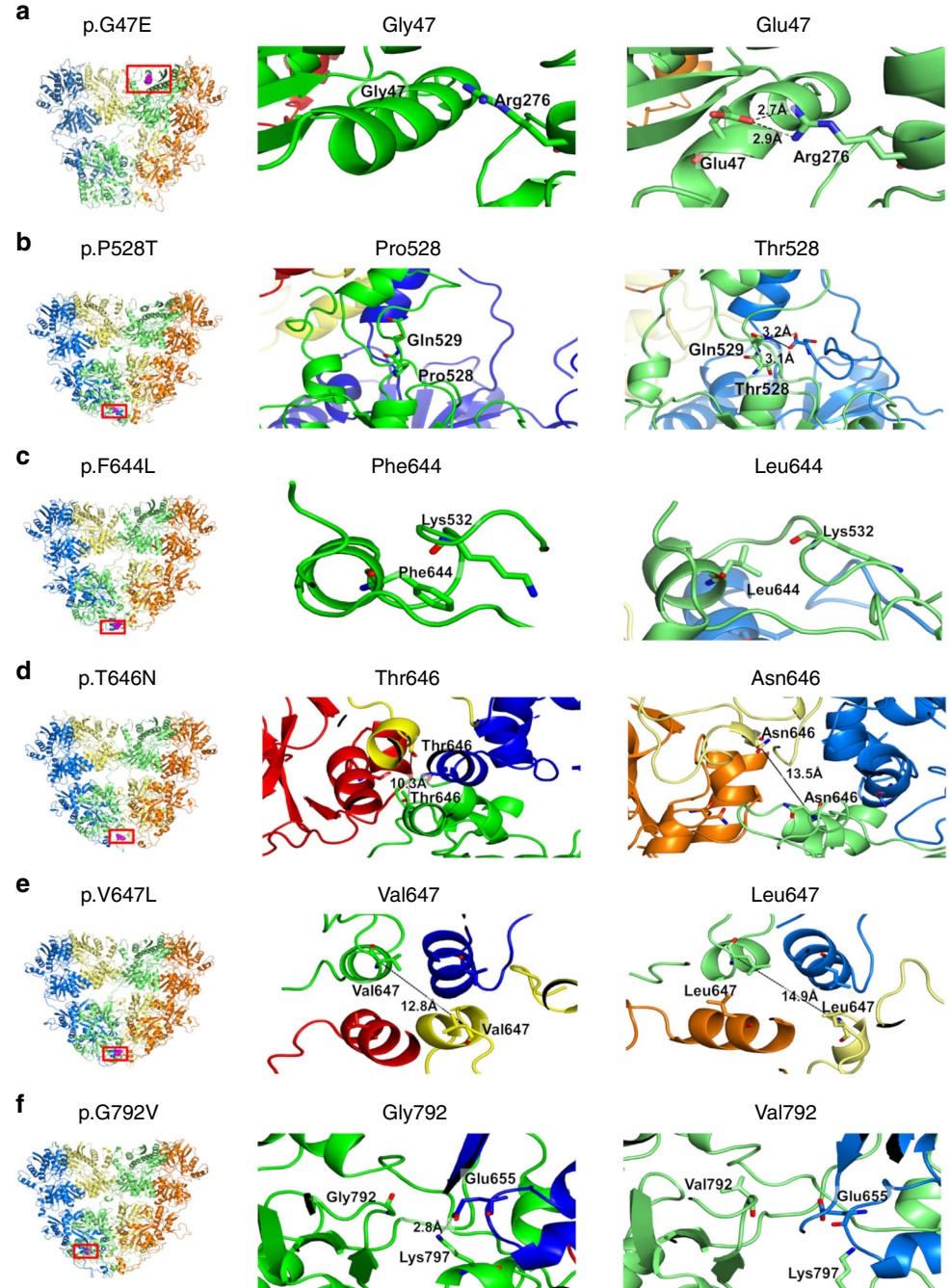

**Fig. 3** Molecular modeling and predicted consequences of 6 *GRIA2* de-novo missense variants Left panel: Six modeled *GRIA2* de-novo missense variants (highlighted in purple within red rectangle) affecting the amino-terminal domain (ATD) and ligand-binding domain (LBD) of GluA2; Middle panel: particular of the wild-type GlUA2 residue; Right panel: particular of the mutated GluA2 residue. **a** For variant p.G47E, the presence of the glutamic acid residue (right panel) in place of the glycine residue (middle panel) is predicted to cause the formation of two hydrogen bonds with a neighboring arginine residue; hydrogen bond distances are shown for the mutant structure. **b** The presence of a Threonine residue in position 528 (right panel) is predicted to cause a slight change in the backbone conformation of the neighboring residues that, in turn, allows hydrogen bonding formation between the two chains forming the LBD domain. **c** Mutation p.F644L causes the loss of hydrophobic interactions between residue 644 and the side chain of Lys532. **d** For mutation p. T646N, the presence of a more hydrophilic Asparagine residue increases the distance between the distal chains by about 3 Å, at the interface of LBD and TMD (right panel). **e** For mutation p.V647L, increased hydrophobicity of Leucine (right panel) compared to Valine (middle panel) increases the separation of helices at the interface between LBD and TMD by about 2 Å. **f** For variant p.G792V, the presence of a more hydrophobic Valine residue in close proximity to the binding site is predicted to cause a sliding movement with respect to the neighboring chain, disrupting the interchain salt bridge between Glu655 and Lys797; hydrogen bond distance is shown in the wild-type GluA2 (middle panel)

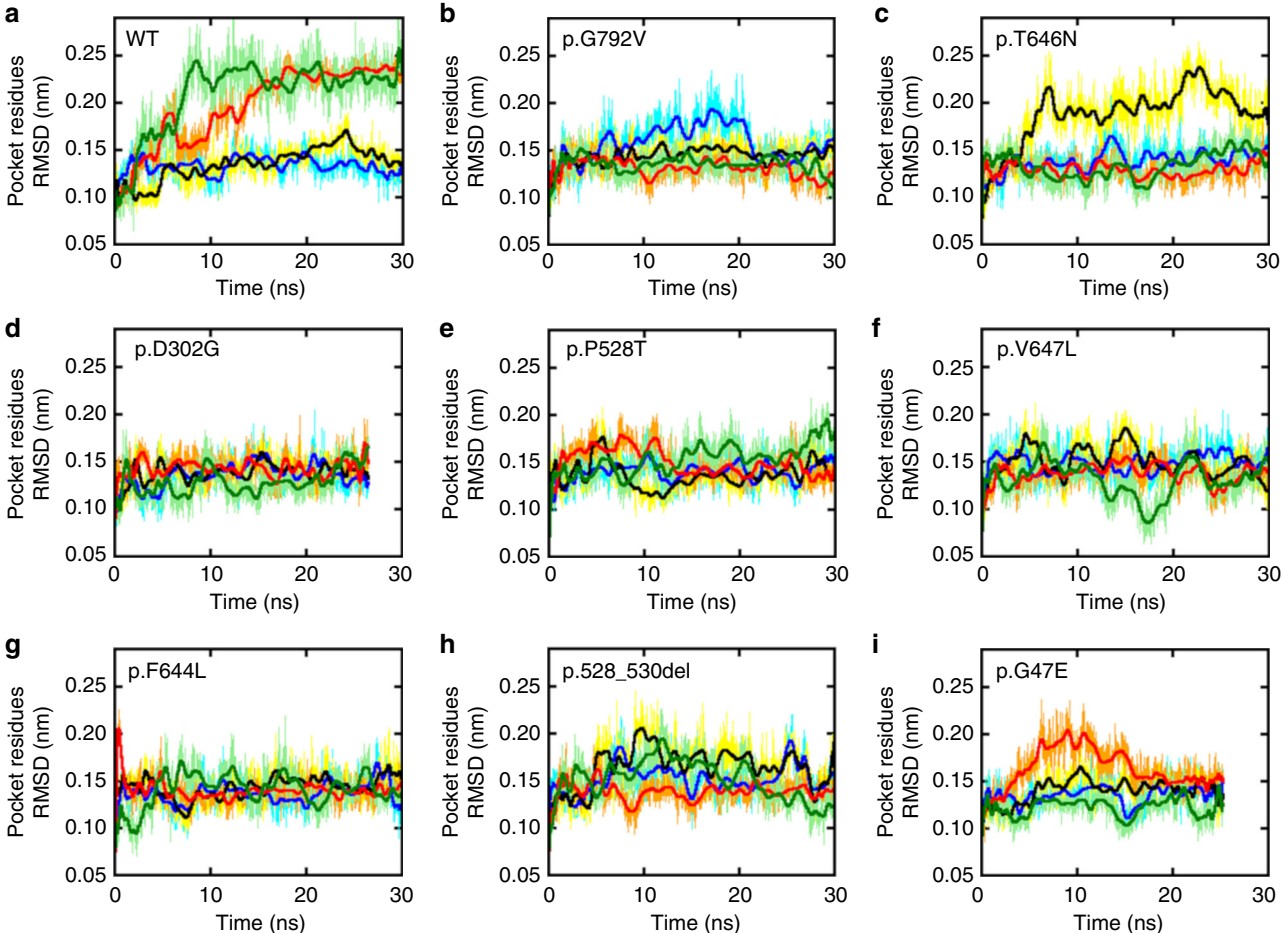

**Fig. 4** GluA2 Molecular dynamic stimulations predict reduced mobility at the agonist binding site. Root Mean Square Deviation (RMSD) of the GluA2 Glutamate binding pocket (GLU) amino acids along the simulated time for the wild-type protein (top left) and *GRIA2* mutants (**a–i**). In all panels the color code is: chain A (blue/cyan), chain B (black/yellow), chain C (red/orange), and chain D (light/dark green). Thicker lines indicate running averages over 100 samples

Although consistent with partial or complete loss of GluA2 incorporation in surface-expressed channels, this does not explain why some mutations that affected rectification, including p. Q607E, did not significantly reduce the KA-evoked current amplitude. Indeed, when voltage ramps were applied to cells expressing p.Q607E without GluA1, the current-voltage relationship was strongly rectifying (Fig. 7), as expected from loss of a polyamine-repelling positively charged residue. Overall, all but one variant (p.P528T) affected either KA-evoked current amplitude or rectification or both when co-expressed with GluA1, although in some cases the effects fell short of significance when unpaired *t*-tests included Holm-Bonferroni correction for multiple comparisons (Fig. 8, Supplementary Table 8).

To determine whether mutations affect channel synthesis or trafficking, we used a biotinylation assay to probe surface expression of selected mutants. When co-expressed with GluA1, p.A639S exhibited a decrease in total expression of GluA2 (Fig. 9, Supplementary Fig. 6). As a fraction of total GluA2, protein at the cell surface was decreased for p.A639S, but also for p.Q607E, consistent with evidence that the arginine normally present at codon 607 affects trafficking[31] (Fig. 9).

We modified the GluA2 sequence to examine the effect of two further codon changes. p.I375V, which is found in 103 subjects out of 60706 individuals from the ExAC database and is therefore likely to be a low-frequency variant of uncertain significance, this increased the current carried by homomeric channels, but not when co-expressed with GluA1 (Supplementary Fig. 7). We also introduced another codon change, p.A643T, that corresponds to the Lurcher mutation in the related non-functional receptor GluD2, because it is near a cluster of 4 mutations identified in the cohort (p.A639S, p.F644L, p.T646N, and p.V647L). The Lurcher mutation disrupts murine cerebellar development and function by creating a leaky receptor that fluxes cations in the absence of ligand. When introduced in GluA2, p.A643T was non-functional, and the holding current was no different from WT-expressing cells (or indeed, cells expressing any of the other variants tested, Supplementary Fig. 7). We conclude that loss of function is caused by multiple molecular mechanisms involving both altered channel surface expression and altered channel function, and that the mechanisms do not involve a Lurcher-like leak conductance (Supplementary Fig. 8).

## Discussion

The genetic and functional expression data presented here identify de-novo mutations and microdeletions involving *GRIA2* as a cause of NDDs and DEE and underline the importance of the GluA2 subunit in the regulation of $Ca^{2+}$ permeation and voltage rectification of AMPARs and therefore in human synaptic plasticity and brain development and function[22,23,28].

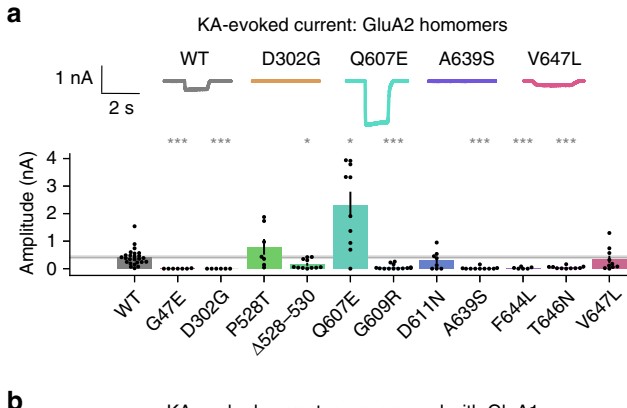

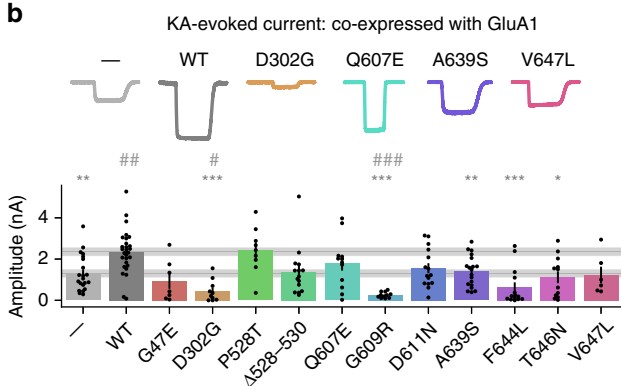

**Fig. 5** Agonist-evoked currents in HEK cells. 1 mM KA was applied transiently to HEK cells expressing GluA2 and stargazin (**a**), or GluA1, GluA2 and stargazin (**b**). Amplitude was compared to WT (*$p < 0.05$, **$p < 0.01$, ***$p < 0.001$) and for co-expression also to the negative control (GluA1 without GluA2, —, #$p < 0.05$, ##$p < 0.01$, ###$p < 0.001$). Mean ± SEM. Data are from the following numbers of independent cells: (**a**) WT:25, G47E:7, D302G:6, P528T:7, Δ528–530:11, Q607E:10, G609R:13, D611N:8, A639S:10, F644L:6, T646N:10, V647L:10 (**b**) —:21, WT:28, G47E:7, D302G:10, P528T:9, Δ528–530:15, Q607E:12, G609R:11, D611N:15, A639S:19, F644L:14, T646N:12, and V647L:6

The structure of each GluA2 subunit includes (i) a large ATD (ii) a LBD formed by the proximal part of the N terminus (S1 lobe) and the large loop between transmembrane segments M3 and M4 (S2 lobe) (Fig. 1), (iii) a TMD formed by hydrophobic membrane-spanning helices M1, M3 and M4 and the M2 helix and re-entrant loop, and (iv) a carboxy-terminal (CTD) intra-cellular region involved in synaptic localization and receptor regulation[30,32]. The intracellular M2 loop together with the M3 helix form the ion-conducting pore[33]. Receptor subunits first form dimers, then tetramers, and structural studies reveal 2-fold symmetry of the extracellular domains which transitions to 4-fold symmetry in the transmembrane domains[34]. Three flexible stretches of amino acids link the ligand-binding domain to the transmembrane helices and allow energy transfer from the ago-nist binding site to the channel gate at the top of M3.

GluA2 subunits are post-transcriptionally edited at the Q/R site at position 607, where M2 protrudes into the pore, rendering the channel non-rectifying and calcium-impermeable. In adults, nearly 100% of GluA2 subunits are in the edited R form. The p. Q607E mutation identified in the present study is caused by a cytosine to guanidine base change immediately 5′ to the adeno-sine that is edited to inosine by ADAR2. The normal editing results in a glutamine (CAG) to arginine (CIG) change, because the inosine base is read as a guanosine by the ribosome. ADAR2 recognition assays predict that the mutant codon (GAG) is edited ~90% less than the normal CAG[35], so we did not investigate the

effect of a glycine (GGG) residue at position 607. When GluA2 was expressed in HEK cells, we observed increased current for Q607E homomers which also exhibited inward rectification. During experiments, we also noticed fewer surviving HEK cells, suggesting a possible toxic effect of this mutation. In contrast, another disease-associated mutation two codons 3′ from codon 607, p.G609R, almost eliminated current in both homomeric and co-expression experiments. Site-directed mutagenesis at or near the Q/R site 607 was previously shown to cause misassembled homomeric GluA2 channels which are retained in the ER[32]. Our data suggest that GluA2 subunits with the G609R mutation are trafficked to the surface but are non-functional and cause co-expressed subunits to also be non-functional. The p.D611N var-iant had a milder effect, with a trend towards decreased function when co-expressed with GluA1.

Importantly, de novo missense variants affecting the M3 channel gate or the M3-S2 linkers have been previously identified in several iGluR subunit genes (e.g., *GRIA1*, *GRIA3*, and *GRIA4*); phenotypes of these patients include ID, autism and epilepsy[7–10]. We identified four mutations associated with neurodevelop-mental phenotypes in on near the SYTANLAAF motif, a highly conserved nine-amino acid region at the top of the M3 trans-membrane helix, which forms the channel gate[36]. Extensive prior work demonstrates the sensitivity of this motif to mutation across the iGluR superfamily. For example, a mutation associated with NDD affecting the eighth residue in GluA1 (p.A636T; SYTAN-LAA**A**F) results in leaky channels[9], and the equivalent mutation (p. A654T) in the *GRID2* gene (encoding the GluD2 receptor) was associated with human movement disorder[37] as well as the Lurcher mouse ataxic phenotype[38]. We tested the analogous GluA2 p.A643T as a positive control in our electrophysiology experiments and observed loss of KA-evoked current but no change in holding current that would suggest a leak. The p.A653T mutation in GluA3, affecting the seventh residue (SYTANL**A**AF), causes NDD and altered sleep and eliminates KA-evoked cur-rents[20]. Structural data in GluA2 homomers places this residue in close proximity to A639 (SYT**A**NLAAF) of the adjacent subunit and suggests that A639 may act as a 'hinge' in the M3 structure[39,40]. In our study, p.A639S caused loss of KA-evoked current in GluA2 homomers and a decrease in current when GluA1was co-expressed, showing that this position is highly sensitive to even a conservative amino acid change. The p.F644L variant is the top of the SYTANLAA**F** motif, and p. T646N and p.V647L map two and three amino acids downstream of the SYTANLAAF motif, respectively. For p.V647L, an inher-ited mutation at the equivalent site in *GRIA1* (p.V640L) caused ASD[9]. For p.T646N, reduced current in GluA1-GluA2 co-expression and partial reduction in RI would be consistent with reduced surface expression combined with impaired channel gating (although a decrease in surface expression was observed it did not reach significance). Molecular dynamic simulations indicated a loss of symmetry between subunit pairs in the pre-sence of p.T646N, with Chain B alone becoming more mobile, in contrast with the WT channel where chains moved in pairs (Fig. 4). This suggests that p.T646N compromises tertiary struc-ture stability. In contrast, p.F644L appeared to be robustly expressed but caused reduction in currents to below the level of GluA1 when co-expressed, demonstrating a gating deficiency which impacts co-assembled WT subunits.

In summary, we noted that effects on currents greatly varied among mutants, with striking differences even when comparing nearby residues. Three mutations (p.D302G, p.G609R, p.F644L) eliminated the GluA1 current when co-expressed, suggesting that they lock other subunits into non-functional channels. They are all located in different domains. Perhaps surprisingly, these mutations are not clearly associated with a more severe

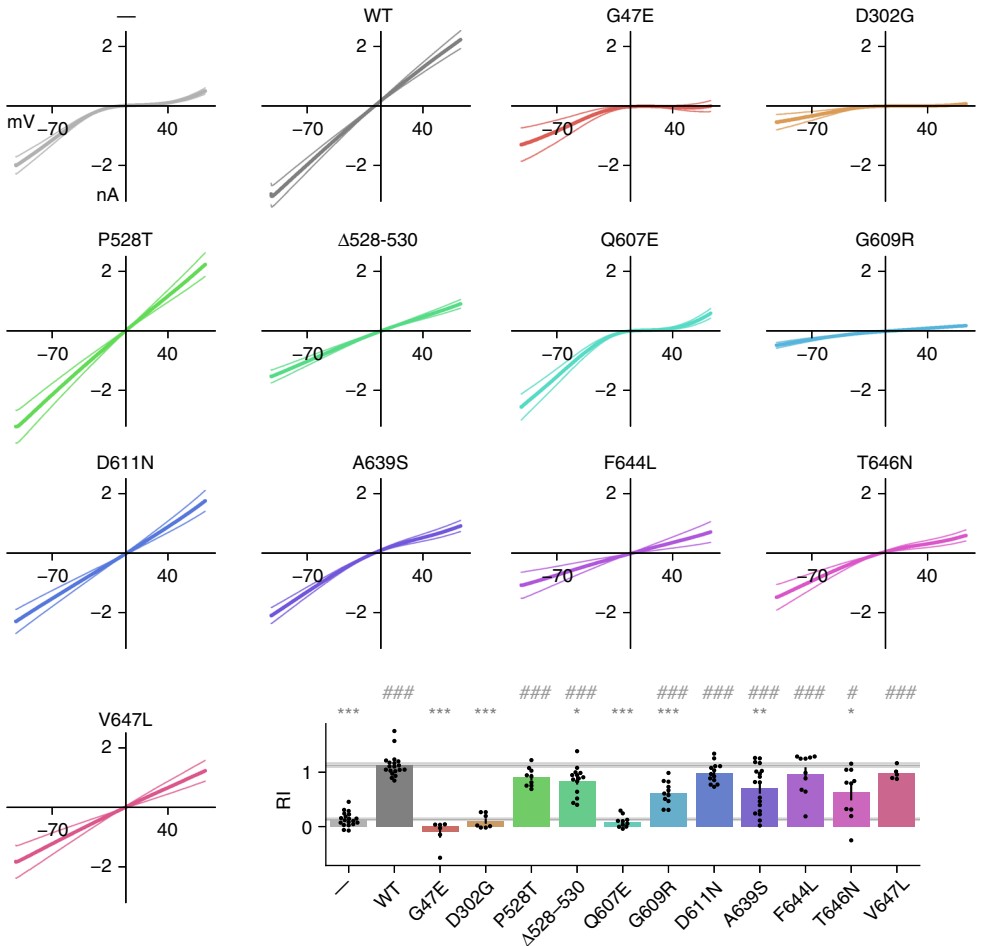

**Fig. 6** Rectification of GluA2 mutants when co-expressed with GluA1. Voltage was ramped from −104 mV to +76 in order to assess rectification of KA-evoked currents. Ramp currents recorded in the absence of KA were subtracted from ramp currents in 1 mM KA. Controls showed linear current–voltage (I–V) relations for cells co-expressing GluA1 and GluA2 WT, and clear rectification in cells transfected with GluA2 alone. The panels show average ramps ±SEM for each mutant ($n \geq 6$ cells per mutant). Rectification index (RI) was quantified as ($I_{+40}/I_{-70}$) *(−7/4). RI was compared to WT (*$p < 0.05$, **$p < 0.01$, ***$p < 0.001$) and GluA1-negative control (#$p < 0.05$, ##$p < 0.01$, ###$p < 0.001$). Data are averaged from the following numbers of cells per mutant: —:19, WT:19, G47E:6, D302G:7, P528T:9, Δ528–530:14, Q607E:10, G609R:11, D611N:13, A639S:18, F644L:10, T646N:10, V647L:5

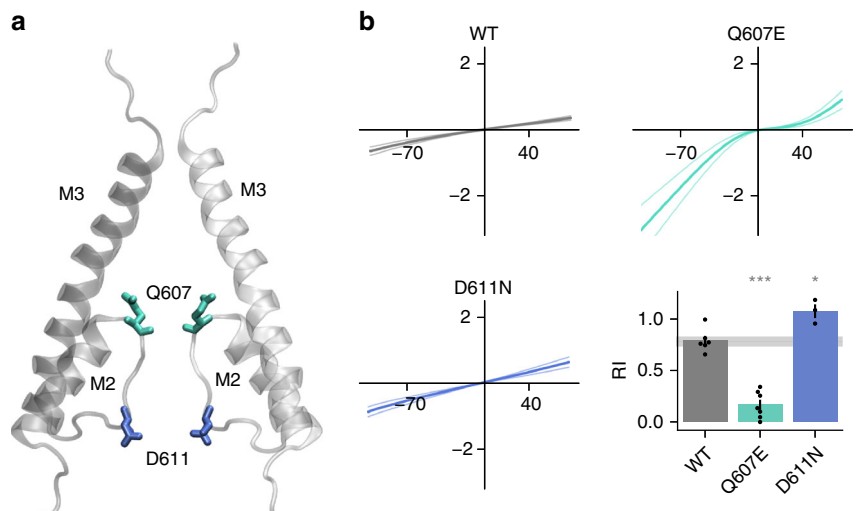

**Fig. 7** Changes at the polyamine binding site alter rectification in GluA2 homomeric channels. **a** Pore detail of 2 opposing subunits in GluA2 tetramer (cryo-EM structure pdb 6dm0). **b** Ramps were applied to HEK cells expressing homomeric GluA2 channels and rectification quantified as for Fig. 4 (*$p < 0.05$, **$p < 0.01$, ***$p < 0.001$). Number of cells recorded per mutant: WT:6, Q607E:7, D611N:3

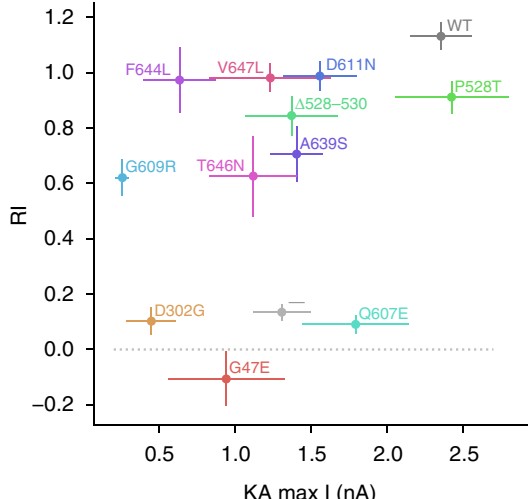

**Fig. 8** Scatter plot summarizing changes in current amplitude and rectification when GluA2 is co-expressed with GluA1 in HEK cells. Relationship between RI and KA-evoked current amplitude for mutant GluA2, compared to WT and negative (GluA1) control (—). Error bars are SEM

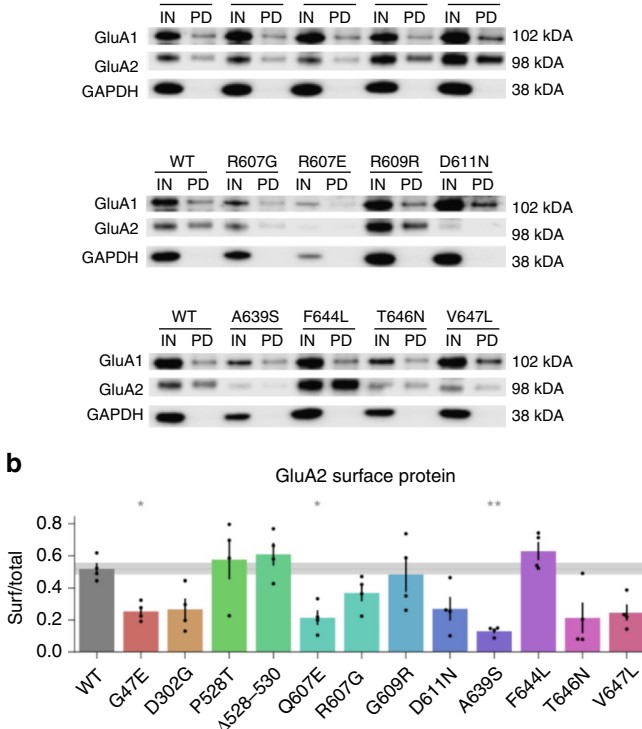

**Fig. 9** Some GluA2 mutations disrupt surface expression. **a** Samples of transfected HEK cells biotinylated before (IN) and after purification by pull down with streptavidin beads (PD). **b** IN and PD were quantified relative to GADPH and then normalized to a WT control on the same gel. Surface protein was quantified as PD/IN. *$p < 0.05$, **$p < 0.01$ vs WT. $N = 4$ experiments from independent batches of transfected cells

phenotype. For another three mutations (p.G47E, p.Q607E, and p.A639S), GluA2 surface protein was significantly reduced in the co-expression western blots, showing that defects in hetero-merization and/or surface trafficking contribute to some pheno-types. This mechanism also shows no correlation with structural

location, again highlighting the diversity among the tested mutants. In general, we are unable to predict clinical severity on the basis of receptor physiology. However, in the case of p.Q607E, we speculate that gain-of-function in the homomeric channel could explain the patient's seizures. The most severe clinical cases were associated with p.A639S, which caused epileptic encepha-lopathy and death in infancy. This mutation caused a significant reduction in total GluA2 protein, but our analysis was hampered by a lack of transfected cells, suggesting that this mutation should be investigated for cellular toxicity.

Further clues to the function of GluA2 come from genetically engineered mice. "Q/R" editing-deficient knock-in mice die of sei-zures at 3 weeks[41], whilst Gria2$^{-/-}$ mice have increased mortality, impaired motor coordination and behavioral abnormalities[28]. Gria2$^{-/-}$ mice also exhibit enhanced long-term potentiation in hippocampal principal neurons that is resistant to blockade of NMDA receptors, consistent with Ca$^{2+}$ entry via AMPARs trig-gering an increase in synaptic strength. A similar NMDAR-independent form of LTP normally occurs in a subset of inter-neurons with rectifying AMPARs, which are deficient in GluA2[42]. Although heterozygous Gria2$^{+/-}$ mice are developmentally normal, they also exhibit NMDAR-independent long-term potentiation in principal neurons, albeit less than homozygous mice[43].

The difference between the severe neurodevelopmental phe-notypes reported in the present study and the heterozygous mice suggests that the human brain is more sensitive to AMPAR dysfunction. For several mutations associated with either NDD or DEE we observed increased rectification of AMPARs when GluA1 and mutant GluA2 were co-expressed, consistent with failure to incorporate mutant GluA2 in heteromeric channels. Although no clear genotype-phenotype correlations emerge for the majority of patients, we did observe a striking correspondence between specific recurrent mutations and the individual pheno-types. Specifically, the p.Val647Leu variant was associated with DEE in 3 cases who showed overlapping electro-clinical features (Supplementary Table 6). Moreover, two individuals (Patient 17, 20) who died with sudden unexplained death in epilepsy (SUDEP) carried the same p.Ala639Ser mutation affecting a conserved Alanine residue proximal to the SYTANLAAF domain. The phenotypic differences in the remaining individuals may arise from different effects of distinct mutations that the in vitro experiments fail to capture. A potential limitation of the present study is that kainate was used as a non-desensitizing agonist. Rapid application of glutamate to outside-out membrane patches may uncover alterations in kinetics that were not captured with kainate application to HEK cells recorded in whole-cell mode. Another possible area for study is the interaction with stargazing and other auxiliary proteins. However, in pilot experiments without stargazin co-expression, no kainate-evoked currents were observed, limiting our ability to quantify this interaction. Further possible contributions to the phenotypes are the effects of mod-ifying genes and stochastic processes during development[44]. The broad range of phenotypes associated in *GRIA2* mutations identified here is reminiscent of the variable neurological phe-notypes reported in association with mutations in other genes encoding homologous AMPAR/NMDAR subunits [7–11,18,20,21]. These include the AMPAR subunit genes *GRIA1*, *GRIA3*, and *GRIA4*, causing an NDD spectrum including ID, loss of speech, epilepsy, gait abnormalities, and abnormal sleep patterns[9,10,20,21].

It is highly likely that dysregulation of a number of tran-scriptional and post-transcriptional modulations is implicated in the *GRIA2* neurodevelopmental disorders. Further studies will determine whether the expression profiles of other genes or proteins contribute to the phenotype associated to *GRIA2* or other AMPAR subunit gene mutations. In patients with *GRIA2*-related disorders, NDD is often associated with a number of

additional clinical features, including neurological and psychiatric comorbidity and other systemic signs, implying that affected individuals should have regular neurological and neuropsychiatric assessments. In a proportion of cases, variants were identified from independent groups of individuals affected with molecularly undefined DEE or RTT-like syndrome (Supplementary notes, Supplementary Tables 5–6). However, WES in our cohort did not identify any pathogenic variants in known NDD- or DEE-associated genes, including *MeCP2* and *CDKL5* which underlie RTT and RTT-like (or DEE) phenotypes, respectively. For patients 26–28, other genes within the 4q32.1 deletion may also contribute to aspects of their clinical phenotype.

Genes associated with DEEs and NDDs that do not encode glutamate receptors may also have an impact on the regulation of excitatory synaptic strength through a variety of mechanisms including *GRIA2* translation and alternative flip/flop splicing[26,27] and abnormal editing at the Q/R site has been also implicated in neurological disorders[45–47]. Taken together, these studies implicate GluA2 dysfunction as a point of convergence for multiple genetic disorders underlying NDD and DEE. Further research is needed to establish the full range of neurological disorders relating to abnormal GluA2 expression and conductance, and to establish whether drugs targeting AMPARs, such as AMPAkines, would ameliorate clinical outcomes. At present, given the evidence for both a decrease in AMPAR function and an increase in $Ca^{2+}$ permeability, we urge caution in extrapolating whether AMPAR inhibitors or desensitization blockers are candidates.

## Methods

**Patients recruitment.** For each affected individual, clinical data as well as brain imaging and EEG were reviewed by the clinicians (geneticists, neurologists, pediatricians) from the participating centers. Genomic DNA was extracted from the whole blood or saliva of the affected individuals and their parents. Informed consent for DNA analysis was obtained from study participants in line with local institutional review board requirements at the time of collection. The study was approved by the ethics committee of University College London (07/Q0512/26) and additional local ethics committees of the participating centers. We complied with all relevant ethical regulations for human patients and obtained informed consents from all the families involved in this study. Families from research participants provided informed consent for publication of the images in Fig. 1b and for publication of videos in the Supplementary Information. Parents of the affected individuals (and when available unaffected siblings) were recruited for segregation analysis, which was carried out using Sanger sequencing. Individuals diagnosed with NDD (including ID, DD, ASD, RTT-like and DEE) were recruited in the different centers participating to the study. Based on the International League against epilepsy (ILAE) classification, a DEE was defined in the patients as refractory seizures and cognitive slowing or regression associated with frequent, ongoing epileptiform activity[48]. Based on the RTT diagnostic criteria[49,50], the affected individuals from this cohort have at least 1/4 main RTT criteria and ≥4 supportive criteria[51,52]. ID was defined based on the presence of significant deficits in conceptual, social and/or practical skills associated with significant deficits in adaptive behavior[53]. Detailed epilepsy and medical histories were obtained together with the results of investigations including EEG and MRI studies. The 28 individuals carrying de-novo *GRIA2* intragenic variants and 4q32.1 microdeletions (Supplementary Figs. 1 and 2, Supplementary Tables 1–4) were recruited from different research groups and consortia in the UK and internationally. Individuals 1, 4, and 17 were studied as part of the SYNAPS Study Group initiative (http://neurogenetics.co.uk/synaptopathies-synaps/). Individuals 3 was initially referred for trio WES to diagnostic laboratories (GeneDX: https://www.genedx.com) from their clinicians from different centers and and followed-up as also part of the SYNAPS Study Group cohort of patients. Also, individuals 6, 7, 8, 21, 22, 24, and 25 were sequenced by trio WES at GeneDX. Individual 2 was initially recruited as part of the DDD Study (DDD4K.03245) and also followed-up in the SYNAPS Study Group. Individuals 10, 11, 12, and 15 were recruited as part of the Autism Clinical and Genetic Resources in China (ACGC)[54] study in China, which consist of more than 4000 individuals affected with ASDs. Patient 19 was recruited in "The Autism Simplex Collection" cohort consisting in 1700 WES trios. Patient 14 was recruited by analyzing the negative exome sequencing data from a published cohort of patients with RTT-like features or DEE[51]. Patient 9 was recruited as part of a project on the genetics of developmental disorders at University Hospital Pitié-Salpêtrière in Paris. Individuals 13, 16, and 18 were recruited within single Institution Epilepsy research centers University Hospital d' enfants Armand Trousseau, University Hospital of Angers, University Hospital of Melbourne) as part of genetic analysis for undiagnosed infantile-onset epileptic

encephalopathies. Individual 5 was recruited at the Leiden University Medical Center as part of a research project on ID and then was submitted to DECIPHER (DECIPHER ID: 322236). Patient 20 was recruited at University Hospital of Sao Paulo. Patient 21 was identified at University of Amsterdam. Patient 22 was identified and genetically investigated at the Center for Autism and Related Disorders in the Kennedy Krieger Institute. Patient 23 was recruited at the Child and Adolescent Psychiatry Unit of the Universidad Complutense in Madrid, Spain. Patient 24 was recruited at Mayo Clinic. Patient 25 was recruited as part of the Care for Rare Program at Ottawa University Children's Hospital in Canada. Individuals 26, 27, and 28 were studied by micro-array analysis as part of single Institutions projects on copy number variants in neurodevelopmental phenotypes and also submitted to DECIPHER (DECIPHER IDs: 328135, 269176, and 296516, respectively). Initial diagnostic work-up (including genetic and metabolic investigations) was normal in all cases. All families gave written informed consent for inclusion in the study and consent for the publication of photographs was obtained for individuals 1, 2, 3, 7, 10, and 12.

**Genetic analyses.** All research centers involved in this study followed a trio-based WES or targeted sequencing approach to identify the de novo *GRIA2* variants as the cause of the neurodevelopmental phenotypes of the patients. The DDD Study analyzed more than 13,000 children with severe developmental disorders and their parents[3], GeneDx laboratory analyzed over than 11,000 individuals affected with NDDs with at least 9000 of them being sequenced with both parents and following the method described above, the SYNAPS Study Group analyzed approximately 260 trios of children with NDDs and EE, the Leiden University Medical Center tested over than 500 ID trios[55]. Following their respective analysis pipelines, participating centers generated a list of candidate variants filtered against public database variants and according to modes of inheritance. All variants reported in the present study were determined independently by participating centers. Connecting the different contributing centers was facilitated by the web-based tools GeneMatcher[31] and DECIPHER[56]. Variants of interest in *GRIA2* gene were mostly identified by WES of trios (Individuals 1, 3, 4, 5, 6, 7, 8, 9, 13, 14, 16, 17, and 18) and targeted sequencing with Molecular Inversion Probes (MIPs, individuals 10, 11, 12, 15, and 19), or Microarray analysis (Individuals 26, 27, and 28). In individuals 3, 6, 7, and 8, trio-based WES was performed at GeneDX using the Clinical Research Exome kit (Agilent Technologies, Santa Clara, CA). Massively parallel (NextGen) sequencing was done on an Illumina system with 100 bp or greater paired-end reads. Reads were aligned to human genome build GRCh37/UCSC hg19, and analyzed for sequence variants using a custom-developed analysis tool[57]. Individuals 10, 11, 12, 15, and 19 were studied through targeted sequencing of ASD candidate genes including *GRIA2* from a cohort of 3910 ASD individuals recruited as part of the ACGC study using a single-molecule molecular inversion probes method[54]. Reads were aligned against hg19 with BWA-MEM (v0.7.13) after removing incorrect read pairs and low-quality reads and single-nucleotide variants and indels were called with Free Bayes (v0.9.14). For Individuals 1 and 17 Nextera Rapid Capture Enrichment kit (Illumina) was used according to the manufacturer instructions. Libraries were sequenced in an Illumina HiSeq3000 using a 100-bp paired-end reads protocol. Sequence alignment to the human reference genome (UCSC hg19), and variants call and annotation were performed using in-house pipelines[39,41]. Libraries were prepared from parent and patient DNA, and exomes were captured and sequenced on Illumina sequencers. Raw data were processed and filtered with established pipelines at the academic or diagnostic laboratories[58–62]. Variant (single nucleotide and indel) calling and filtering was performed using the Genome Analysis Tool Kit (GATK; see URLs). Variants that did not adhere to the following criteria were excluded from further analysis: allele balance of >0.70, QUAL of <20, QD of <5, and coverage of <20×. Variants were annotated and the Exome variant server ESP6500 (see URLs) was used to assess variant frequency in the control population. In the index case (Individual 1) trio WES, the average sequencing depth of the on-target regions was 76.8 reads per nucleotide, with 96.8% of the regions covered at least 20×. In our analysis, we excluded non-exonic variants and exonic synonymous variants and prioritized rare variants (with a frequency <1% in ExAC and 1000 Genomes project). Traditional Sanger sequencing was used to validate the variants and to assess their segregation within the families (detailed conditions of the primers used, and sequencing methods are available upon request). In regard to variants filtering and interpretation, autosomal recessive and dominant de-novo mutations were prioritized in our analysis at the different centers. Variants were annotated using the Variant Effect Predictor (Ensembl release 75) based on Sequence Ontology nomenclature: missense variant, initiator codon variant, splice donor or acceptor variant, frameshift variant, stop lost, stop gained, in-frame insertion or deletion. We prioritized annotations using the transcript associated with the most severe consequence for each variant and, in case of similar consequences, we prioritized the flip transcript (NM000826.3) being the one with largest base pairs length. To exclude likely benign amino acid changes, non-synonymous variants were further considered if predicted damaging by at least 3 out of 5 in-silico methods among PolyPhen-2, SIFT, Mutation Taster, Condel and CADD (see URLs). Variants that were not present in both the mother and the father of the probands were considered de-novo. In recessive filtering, we included homozygous, hemizygous or compound heterozygous variants. Variants present in >1% of our internal exome dataset at the UCL Institute of Neurology (containing ~5000 exomes from

individuals affected with a range of neurological disorders) were excluded. Exome data were analyzed for variants in genes linked before to RTT or RTT-like syndrome, epilepsy and NDDs, and for variants in other genes not linked to diseases. Genes involved in EE and RTT-like presentations were retrieved from the literature[58]. Based on values from the ExAC database (containing 60,706 individuals), variants in genes with high probability of being LoF intolerant (i.e., ExAC pLI >0.9) and highly constrained for missense variations (Z-score >2) were prioritized. In the case of candidate genes, variants in genes whose homologs are known to be implicated in neurological and neurodevelopmental disorders were prioritized in the analysis. Patient was analyzed in the discovery phase of this study and found to carry a single de-novo exonic variant (with a MAF <0.001) in *GRIA2* (NM001083619.1: c.2363G>T; p.Trp788Leu). This was confirmed by Sanger in the trio. Similarly to the index case (Patient 1), also in other research and diagnostic laboratories the identified variants in *GRIA2* were prioritized and emerged as the most likely explanation for the individuals disease pathogenesis, as supported by (i) high conservation of the affected residue across species, as well as in-silico analysis and high pathogenic scores (Supplementary Table 1); (ii) biological importance of the residues affected by the mutations (the identified variants mostly affect conserved sites within the transmembrane domain known to be important in GluA2 and AMPAR function); (iii) crucial function of the gene and its encoded protein in synaptic plasticity and brain development and function; (iv) publications linking this gene homologs (*GRIA1*, *GRIA3*, *GRIA4*) to similar NDD phenotypes; (v) de-novo occurrence of the *GRIA2* variants which was demonstrated in all the laboratories by trio-based traditional Sanger sequencing. The comparison of phenotypes across the different *GRIA2* mutated individuals identified within the different centers involved the study confirmed the implication of *GRIA2* de-novo variants in the observed spectrum of neurological abnormalities.

**Functional characterization of the identified *GRIA2* variants.** Human GluA2 (flip, Q/R edited) plasmids were produced under contract by Genscript, USA. cDNA was synthesized and cloned into pcDNA3.1+ using HindIII and XhoI, prior to mutagenesis. pIRES2-GFP-Stargazin was a gift from Stuart Cull-Candy and Mark Farrant, University College London. HEK cells were cultured in DMEM with 10% FBS and passaged 2 times per week. Cell line verification was not carried out, however all experiments used the same frozen stocks and mutant experiments were interleaved with controls. Twenty-four hours prior to transfection, HEK cells were seeded in 6-well plates at a cell density of 300,000 cells/well in 2 mL media. For western blots the wells were pre-coated with poly-D-lysine. Cells were transfected using TurboFect (ThermoFisher, UK) according to the manufacturer's protocol. For experiments testing homomeric channels, 1.5 μg GluA2 (WT or mutant) plasmid was combined with 1 μg stargazin-GFP plasmid. Negative controls for western blots excluded GluA2. For experiments testing co-expressed GluA1 and GluA2 (WT or mutant), both plasmids and stargazin were co-transfected at 0.7 μg each. GluA1 without GluA2 acted as the negative control. Transfection proceeded for 4–6 h, and then cells were washed (for Western blots) or re-plated onto poly-D-lysine coated 13 mm round coverslips (for electrophysiology).

Cells were perfused with external solution composed of: (mM): NaCl 140, KCl 2.4, CaCl$_2$ 2, MgCl$_2$ 1, HEPES 10, Glucose 10; pH 7.4 (NaOH). Internal solution was composed of (mM): CsCl 145, CaCl$_2$ 2, MgCl$_2$ 2, EGTA 10, HEPES 10, Glucose 17.5; pH 7.4 (CsOH). Patch pipettes were typically of 4.5MΩ resistance after polishing. Cells were whole-cell voltage clamped at −74 mV (adjusted for a liquid junction potential of 4 mV). Series resistance did not exceed 25 MΩ and compensation was not applied. Data were acquired with a Multiclamp 700B (Molecular Devices, USA), logged at 1 kHz using a BNC-2090A (National Instruments, USA) and WinEDR version 3.8.0 (University of Strathclyde, UK). Cells on coverslips were secured in a custom-made bath and visualized by GFP fluorescence on an IX73 inverted microscope (Olympus). Kainic acid (Hello Bio, UK) was diluted freshly on the day of recording at 1 mM. Control and drug solutions were applied to a patch-clamped cell in parallel streams through a glass theta tube (TGC150–10, Harvard Apparatus), pulled with openings ~300 μm wide. We used a hand-operated piezo manipulator (Scientifica) to switch solutions, giving exchange times of ~100 ms. Where GluA1 and GluA2 were co-expressed, incorporation of both subunits was probed with voltage ramps from −104 to + 76 mV, over 1.8 s. Peak current amplitude was measured using WinEDR. Ramp currents during kainate application were analyzed using a custom Python script.

For the biotinylation assay, the cells were washed with buffer PBS containing 0.1 mM CaCl$_2$ and 1 mM MgCl$_2$ (PBS-CM) and incubated with 600 μl of the same buffer containing 0.5 g/ml of sulfo-NHS-Biotin (Thermo Fisher) for 30 min on ice. The cells were then washed with 100 mM glycine in PBS-CM and incubated in the same buffer 20 min on ice. After that, cells were washed twice with PBS and lysate with 300 μl of lysis buffer composed of: 100 mM NaCl, 5 mM EDTA, 1% (v/v) Triton X-100, protease inhibitor cocktail (Sigma) and 50 mM HEPES, pH 7.4 (pH 7.4). Remained cells were scraped and then sonicated for 5 min in a 0.5 ml microcentrifuge tube and vortexed 15 min at RT. Then, the lysate was centrifuged 10 min at 20,000 × g 4 °C. Aliquots of 15 μl of supernatant were stored as "input controls" and the remaining supernatant was added to 40 μl of Neutravidin-agarose beads (Thermo Fisher) previously washed twice with lysis buffer, and incubated on a rotor wheel for 1 h at RT. Samples were then eluted with 4x LDS sample buffer for 20 min at 76 °C. Input (lysate, 1.25% of total) and eluates (33% of total) were subjected to SDS–PAGE (Bis–Tris 4–12% gradient gels, Invitrogen) in MES buffer

(Invitrogen), and transferred to PVDF membranes. The membranes were immunoblotted whit primary antibodies anti-GluA1, anti-GluA2 (Alomone Labs, #cat: AGC-004 and AGC-005, concentration 1:500) and anti-GAPDH (Abcam, #cat: ab9483, concentration 1:5000) over night and secondary HRP antibody for 30 min. Quantification was performed using Bio-Rad Image Lab. Reproducibility of results was confirmed by performing three independent experiments.

**Statistics.** Statistics were performed in Python version 3.6.1 (Anaconda version 4.4.0) with scipy version 0.19.0 and statsmodels version 0.8.0. Data for mutant channels were compared with WT using independent *t*-tests in scipy.stats, without assuming equal variances (Welch's correction). *P*-values were then adjusted for multiple comparisons using statsmodels.sandbox.stats.multicomp.multipletests to apply step-down Holm-Bonferroni *p*-value adjustments. The *p*-value adjustment was performed on the whole data set simultaneously, including the mutations presented in Supplementary Figs. 6–8. All figures show mean ± SEM.

**Molecular modeling and dynamic stimulations.** The soluble WT AMPA fragment was built from structure PDB ID 3KG2[59]. Missing atoms were added with DeepView - Swiss-PdbViewer 4.1[60] and removed its transmembrane fragment 514–617 and 789–817 identified with PPM server http://opm.phar.umich.edu. The resulting model (comprising fragments 10–513 and 618–788) was minimized, then placed in a cubic box with a water layer of 0.7 nm and Na$^+$ Cl$^−$ ions to neutralize the system, and a second minimization was performed. The resulting structure was then employed as a template ad was mutated to obtain all the constructs using the software Coot[61] for mutation, rotamer manual selection and regularization of the backbone. All models were subsequently minimized, placed once again in a cubic box with a water layer of 0.7 nm and Na + Cl− ions to neutralize the system, and a second minimization was performed. In all cases we used AMBER99SB-ILDN[62] force field and Simple Point Charge water. On all systems we performed NVP and NPT equilibrations for 100 ps, followed by 30 ns NPT production run at 300 K. The temperature was controlled with a modified Berendsen thermostat[63], the pressure with an isotropic Parrinello-Rahman at 1 bar. The iteration time step was set to 2 fs with the Verlet integrator and LINCS constraint[64]. We used periodic boundary conditions. Configurations were sampled every 10 ps. All the simulations and their analysis were run as implemented in the GROMACS package[65]. Figures showing details of the molecular structures of AMPA GLU and its mutants were made with PyMOL (www.pymol.org). To compare the structural mobility of GluA2 and its mutants we built a model of each protein ectodomain including LBD and ATD by modeling mutations on the wild-type and followed their behavior along time by means of atomistic molecular dynamics simulations in water solvent. Proteins with mutations in the pore region where omitted as either the mutations were close to or included in the TMDs which were not modeled.

**URLs.** For Interactive bio-software, see https://www.interactive-biosoftware.com/doc/alamut-visual; for CADD, see http://cadd.gs.washington.edu/; for ClustalX, see http:// www.ebi.ac.uk/Tools/msa/clustalw2/; for Exome Variant Server of the National Heart, Lung, and Blood Institute Grand Opportunity (NHLBI GO) Exome Sequencing Project (accessed February 2014), see http://evs.gs.washington.edu/EVS/; for Genome Analysis Toolkit (GATK), see http://www.broadinstitute.org/gatk/; for GenotypeTissue Expression (GTEx) Project, see http://www.gtexportal.org/; for NCBI ClinVar database, see http://www.ncbi.nlm.nih.gov/clinvar/; for Online Mendelian Inheritance in Man (OMIM), see http://omim.org/; for Picard, see http://broadinstitute.github.io/ picard/; for Primer-BLAST, http://www.ncbi.nlm.nih.gov/tools/primerblast/; for UCSC Genome Browser, see http://genome.ucsc.edu/; for UniProt database, see http://www.uniprot.org/; for Exome Variant Server, see evs.gs.washington.edu/; for Ensembl, see https://www.ensembl.org/; for GnomAD, see http://gnomad.broadinstitute.org/; for GTEx, see https://www.gtexportal.org/home/; for Exome Aggregation Consortium (ExAC), see www.exac.broadinstitute.org; for LOVD, see https://www.lovd.nl.

**Reporting summary.** Further information on research design is available in the Nature Research Reporting Summary linked to this article.

# Data availability

The authors declare that all the data supporting the findings of this study are included in the article (or in the Supplementary material) and available from the corresponding author (H.H.). The source data underlying Figs. 3, 4, 5, 6, and 7 are provided as Source Data files (https://figshare.com/s/9bd6a3ebb2f304d31b59). Data of mutations reported within this study have been deposited in Leiden Open Variation Database (accession numbers for the DNA sequences: 00231337; 00231345; 00231346; 00231347; 00231348; 00231349; 00231388; 00231389; 00231356; 00231359; 00231362; 00231363; 00231365; 00231366; 00231368; 00231369; 00231371; 00231372; 00231376; 00231377; 00231378; 00231379; 00231380; 00231381).

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

## Acknowledgements

We gratefully acknowledge all the families for their enthusiastic participation to this study. Further acknowledgements can be found in Supplementary Note 29. This study was supported by the Wellcome Trust (WT093205MA and WT104033AIA), Medical Research Council (H.H. and D.M.K.), European Community's Seventh Framework Programme (FP7/2007-2013, under grant agreement No. 2012-305121 to H.H.), Muscular Dystrophy Association (MDA), Muscular Dystrophy UK, The MSA Trust, Ataxia UK, The Sparkes Children's Medical Research Charity, The Great Ormond Street Hospital Charity, Rosetrees Trust, Brain Research UK, The UK HSP Society, The European Union's Horizon 2020 research and innovation programme Solve-RD project (No 779257), The Pakistan Council (Scholarship to HT), The National Natural Science Foundation of China (31671114, 81871079, 81330027, and 81525007 to H.G. and K.X.) and the US National Institutes of Health (NIH grant R01MH101221 to E.E.E.). E.E.E. is an investigator of the Howard Hughes Medical Institute. We acknowledge the CINECA Awards N. HP10BTJPER, 2017 (to SF), for the availability of high performance computing resources and support. We are also supported by the National Institute for Health Research (NIHR) University College London Hospitals (UCLH) Biomedical Research Centre (BRC). We are grateful to M. Farrant and S. Cull-Candy for helpful suggestions and the stargazin plasmid. We also acknowledge the University of Washington Center for Mendelian Genomics.

## Author contributions

V.S., C.L.D., H.G., E.E.E., D.M.K., and H.H. coordinated the study. E.T and M.T.C. participated in the recruitment of patients 3, 6, 7, 8, 21, 22, 24, and 25. K.G.M., T.S.S., R.E.P., R.W., and M.T.C. were involved in the analysis and interpretation of the clinical and genetic from patients 3, 6, 7, and 8, 22, 24, 25. G.H., O.B.Y., M.T., and B.B. ascertained and recruited Patient 17. L.B. and S.V. ascertained and recruited Patient 13. O.D.B. and M.H. performed the biotinylation assays and participated in the analysis of results and in manuscript revision. H.T. and S.E. participated in the analysis of the results from DNA sequencing and performed Sanger sequencing and mutagenesis. E.C., V.P., S.G., and D.B. ascertained and recruited Patient 18 and also participated in manuscript revision. P.S., F.Z., S.G., and C.M. screened a replication cohort, participated in patients phenotyping, and contributed to the revision. K.M. reviewed the brain imaging of the patients. A.L. participated in the electrophysiology experiments and/or in the analysis of the electrophysiology results. J.R. participated in the follow-up of Patient 2 and ascertained and recruited Patient 20. S.C. and L.P. ascertained and recruited Patient 8. A.V.H. and C.R. ascertained and recruited Patient 5. C.N., D.H., and B.K. ascertained and recruited Patient 9. R.D.Z. and S.F. modeled the variants and performed and analyzed molecular dynamic stimulations. M.C. ascertained and recruited Patient 7. A.L.S., A.M. M., K.B., H.C.M., and I.E.S. recruited and followed-up Patient 16 and participated in manuscript revision and replication cohort screening. A.B., E.G., and G.B.F. recruited and followed-up patient 22. H.G., Y.Q., H.W., T.W., R.A.B., and KX recruited and followed-up patients 10, 11, 12, 15, and 19 and participated to genetic analysis. R.M. and M.I. participated in the revision of the manuscript, the revision of the references and the formatting. A.T. was involved in the phenotyping and clinical follow-up of Patient 3. A.B. and G.B.F. ascertained and recruited Patient 20. E.F., M.S., and J.C. ascertained and recruited Patient 6. C.B., R.M., and J.V. participated in the sequencing analysis results and in the preparation of the revised ì. M.C., Y.Y., and J.H.C. ascertained and recruited Patient 14. J.C.S. ascertained and recruited Patient 4. J. M.R.C. and A.M. recruited and followed-up Patient 1. M.P., C.L., J.G.P., A.C. recruited and ascertained Patient 23. D.D. and M.O. recruited and ascertained Patient 25. F.K. ascertained and recruited Patient 20. M.M.M. and B.J. recruited and ascertained Patient 21. L.S.R., E.W.L, R.G. and L.G. recruited and ascertained Patient 24. A.B. and M.J. ascertained and recruited Patient 22. M.P. and A.C. ascertained and recruited Patient 23. R.H.G., L.G., L.S.R., and E.W.K ascertained and recruited Patient 24. C.L.D. performed and analyzed the results of electrophysiology. D.M.K. analyzed the results of electrophysiology experiments and revised the manuscript. H.H. coordinated the study revised the manuscript. V.S. and C.L.D. wrote the initial draft of the manuscript, with contributions and/or revisions from all authors.

## Additional information

**Competing interests:** E.T., K.G.M., T.S.-S., R.E.P., R.W., and M.T.C. are employees of GeneDx. E.E.E. is on the scientific advisory board (SAB) of DNAnexus, Inc. The remaining authors declare no competing interests.

Vincenzo Salpietro[1,2,3,140], Christine L. Dixon[4,140], Hui Guo[5,6,140], Oscar D. Bello[4,140], Jana Vandrovcova[1], Stephanie Efthymiou[1,4], Reza Maroofian[1], Gali Heimer[7], Lydie Burglen[8], Stephanie Valence[9], Erin Torti[10], Moritz Hacke[11], Julia Rankin[12], Huma Tariq[1], Estelle Colin[13,14], Vincent Procaccio[13,14], Pasquale Striano[2,3], Kshitij Mankad[15], Andreas Lieb[4], Sharon Chen[16], Laura Pisani[16], Conceicao Bettencourt[17], Roope Männikkö[1], Andreea Manole[1], Alfredo Brusco[18], Enrico Grosso[18], Giovanni Battista Ferrero[19], Judith Armstrong-Moron[20], Sophie Gueden[21], Omer Bar-Yosef[7], Michal Tzadok[7], Kristin G. Monaghan[10], Teresa Santiago-Sim[10], Richard E. Person[10], Megan T. Cho[10], Rebecca Willaert[10], Yongjin Yoo[22], Jong-Hee Chae[23], Yingting Quan[6], Huidan Wu[6], Tianyun Wang[5,6], Raphael A. Bernier[24], Kun Xia[6], Alyssa Blesson[25], Mahim Jain[25], Mohammad M. Motazacker[26], Bregje Jaeger[27], Amy L. Schneider[28], Katja Boysen[28], Alison M. Muir[29], Candace T. Myers[30], Ralitza H. Gavrilova[31],

Lauren Gunderson[31], Laura Schultz-Rogers[31], Eric W. Klee[31], David Dyment[32], Matthew Osmond[32,33,34],
Mara Parellada[35], Cloe Llorente[36], Javier Gonzalez-Peñas[37], Angel Carracedo[38,39], Arie Van Haeringen[40],
Claudia Ruivenkamp[40], Caroline Nava[41], Delphine Heron[41], Rosaria Nardello[42], Michele Iacomino[43],
Carlo Minetti[2,3], Aldo Skabar[44],
Antonella Fabretto[44], SYNAPS Study GroupMiquel Raspall-Chaure[45], Michael Chez[46], Anne Tsai[47],
Emily Fassi[48], Marwan Shinawi[48], John N. Constantino[49], Rita De Zorzi[50], Sara Fortuna[50], Fernando Kok[51,52],
Boris Keren[41], Dominique Bonneau[13,14], Murim Choi[22], Bruria Benzeev[7], Federico Zara[43],
Heather C. Mefford[29], Ingrid E. Scheffer[28], Jill Clayton-Smith[53,54], Alfons Macaya[45], James E. Rothman[4,55],
Evan E. Eichler[5,56], Dimitri M. Kullmann[4] & Henry Houlden[1]

[1]Department of Neuromuscular Disorders, UCL Queen Square Institute of Neurology, London WC1N 3BG, UK. [2]Pediatric Neurology and Muscular Diseases Unit, IRCCS Istituto "Giannina Gaslini", 16147 Genoa, Italy. [3]Department of Neurosciences, Rehabilitation, Ophthalmology, Genetics, Maternal and Child Health, University of Genoa, 16132 Genoa, Italy. [4]Department of Clinical and Experimental Epilepsy, UCL Queen Square Institute of Neurology, London WC1N 3BG, UK. [5]Department of Genome Sciences, University of Washington School of Medicine, Seattle, Washington 98195, USA. [6]Center for Medical Genetics & Hunan Key Laboratory of Medical Genetics, School of Life Sciences, Central South University, Changsha 410083 Hunan, China. [7]Pediatric Neurology Unit, Safra Children's Hospital, Sheba Medical Center and Sackler Faculty of Medicine, Tel Aviv University, Tel Aviv 526121 Ramat Gan, Israel. [8]Centre de Référence des Malformations et Maladies Congénitales du Cervelet, Département de Génétique et Embryologie Médicale, APHP, Hôpital Trousseau, 75012 Paris, France. [9]Centre de Référence des Malformations et Maladies Congénitales du Cervelet, Service de Neurologie Pédiatrique, APHP, Hôpital Trousseau, 75012 Paris, France. [10]GeneDx, Gaithersburg, MD 20877, USA. [11]Biochemistry Center, Heidelberg University, D-69120 Heidelberg, Germany. [12]Royal Devon and Exeter NHS Foundation Trust, Exeter EX1 2ED, UK. [13]Department of Biochemistry and Genetics, University Hospital, 49933 Angers, France. [14]MitoLab, UMR CNRS 6015-INSERM U1083, MitoVasc Institute, Angers University, 49100 Angers, France. [15]Great Ormond Street Hospital for Children, London WC1N 3JH, UK. [16]Division of Medical Genetics, Northwell Health/Hofstra University SOM, New York 11020, USA. [17]Department of Clinical and Movement Neurosciences and Queen Square Brain Bank for Neurological Disorders, UCL Queen Square Institute of Neurology,, London WC1N 1PJ, UK. [18]Department of Medical Sciences, Medical Genetics Unit, University of Torino, 10126 Torino, Italy. [19]Department of Public Health and Pediatrics, University of Torino, 10126 Torino, Italy. [20]Unit of Medical and Molecular Genetics, University Hospital Sant Joan de Deu Barcelona, 08950 Barcelona, Spain. [21]Unit of Neuropediatrics, University Hospital, Angers Cedex 49933, France. [22]Department of Biomedical Sciences, Seoul National University, Seoul 03080, South Korea. [23]Department of Pediatrics, Seoul National University, Seoul 03080, South Korea. [24]Department of Psychiatry, University of Washington, Seattle, WA 98195, USA. [25]Center for Autism and Related Disorders, Kennedy Krieger Institute, Baltimore, Maryland 21211, USA. [26]Department of Clinical Genetics, University of Amsterdam, Meibergdreef 9, 1105 Amsterdam, Netherlands. [27]Department of Pediatric Neurology, Amsterdam UMC, 1105 Amsterdam, Netherlands. [28]Epilepsy Research Centre, Department of Medicine, University of Melbourne, Austin Health, Melbourne, Victoria 3084, Australia. [29]Department of Pediatrics, University of Washington, Seattle, WA 98195, USA. [30]Department of Pediatrics, Division of Genetic Medicine, University of Washington, Seattle, WA 98195, USA. [31]Department of Clinical Genomics, Mayo Clinic, Rochester 55902 MN, USA. [32]Children's Hospital of Eastern Ontario Research Institute, University of Ottawa, Ottawa K1H 8L1, Canada. [33]Department of Human Genetics, McGill University Health Centre, Montréal, QC H4A 3J1, Canada. [34]Genome Québec Innovation Center, Montréal, QC H3A 0G1, Canada. [35]Child and Adolescent Psychiatry Department, Hospital General Universitario Gregorio Marañón, School of Medicine, Universidad Complutense, IiSGM, CIBERSAM, 28007 Madrid, Spain. [36]Institute of Psychiatry and Mental Health, Hospital General Universitario Gregorio Maranon, Universidad Complutense, CIBERSAM, 28007 Madrid, Spain. [37]Hospital Gregorio Maranon, IiSGM, School of Medicine, Calle Dr Esguerdo, 46, 28007 Madrid, Spain. [38]Grupo de Medicina Xenómica, Centro de Investigación Biomédica en Red de Enfermedades Raras (CIBERER), CIMUS, Universidade de Santiago de Compostela, 15782 Santiago de Compostela, Spain. [39]Fundación Pública Galega de Medicina Xenómica- IDIS- Servicio Galego de Saúde (SERGAS), 15706, 15782 Santiago de Compostela, Spain. [40]Department of Clinical Genetics, Leiden University Medical Center, 2333 ZA Leiden, Netherlands. [41]Department of Genetics, Assistance Publique - Hôpitaux de Paris, University Hôpital Pitié-Salpêtrière, 75013 Paris, France. [42]Department of Health Promotion,Mother and Child Care, Internal Medicine and Medical Specialities "G. D'Alessandro", University of Palermo, 90133 Palermo, Italy. [43]Laboratory of Neurogenetics and Neuroscience, IRCCS Istituto "Giannina Gaslini", 16147 Genova, Italy. [44]Institute for Maternal and Child Health, IRCCS "Burlo Garofolo", University of Trieste, 34134 Trieste, Italy. [45]Department of Pediatric Neurology, University Hospital Vall d'Hebron, Universitat Autònoma de Barcelona, 08035 Barcelona, Spain. [46]Neuroscience Medical Group, 1625 Stockton Boulevard, Suite 104, Sacramento, CA 95816, USA. [47]Department of Genetics and Inherited Metabolic diseases, Children's Hospital Colorado, Aurora, CO 80045, USA. [48]Department of Pediatrics, Washington University School of Medicine, St. Louis, MO 63110, USA. [49]William Greenleaf Eliot Division of Child & Adolescent Psychiatry, Department of Psychiatry, Washington University School of Medicine, St. Louis, MO 63110, USA. [50]Department of Chemical and Pharmaceutical Sciences, University of Trieste, 34134 Trieste, Italy. [51]Neurogenetics Unit, Department of Neurology, University of Sao Paulo, Sao Paulo 01308-000, Brazil. [52]Mendelics Genomic Analysis, Sao Paulo, SP 04013-000, Brazil. [53]Centre for Genomic Medicine, Manchester Academic Health Sciences Centre, Central Manchester University Hospitals NHS Foundation Trust, Lancashire M13 9WL, UK. [54]Division of Evolution and Genomic Sciences, School of Biological Sciences, University of Manchester, Manchester M13 9WL, UK. [55]Department of Cell Biology, Yale University School of Medicine, New Haven, CT 06520, USA. [56]Howard Hughes Medical Institute, University of Washington, Seattle, WA 98195, USA. [140]These authors contributed equally: Vincenzo Salpietro, Christine L. Dixon, Hui Guo, Oscar D. Bello. A full list of consortium members appears at the end of the paper.

## SYNAPS Study Group

Michael G. Hanna[1], Enrico Bugiardini[1], Isabel Hostettler[1], Benjamin O'Callaghan[1], Alaa Khan[1], Andrea Cortese[1],
Emer O'Connor[1], Wai Y. Yau[1], Thomas Bourinaris[1], Rauan Kaiyrzhanov[1], Viorica Chelban[1], Monika Madej[1],

Maria C. Diana[2], Maria S. Vari[2], Marina Pedemonte[2], Claudio Bruno[2], Ganna Balagura[3], Marcello Scala[3], Chiara Fiorillo[3], Lino Nobili[3], Nancy T. Malintan[4], Maria N. Zanetti[4], Shyam S. Krishnakumar[4], Gabriele Lignani[4], James E.C. Jepson[4], Paolo Broda[43], Simona Baldassari[43], Pia Rossi[43], Floriana Fruscione[43], Francesca Madia[43], Monica Traverso[43], Patrizia De-Marco[43], Belen Pérez-Dueñas[45], Francina Munell[45], Yamna Kriouile[57], Mohamed El-Khorassani[57], Blagovesta Karashova[58], Daniela Avdjieva[58], Hadil Kathom[58], Radka Tincheva[58], Lionel Van-Maldergem[59], Wolfgang Nachbauer[60], Sylvia Boesch[60], Antonella Gagliano[61], Elisabetta Amadori[62], Jatinder S. Goraya[63], Tipu Sultan[64], Salman Kirmani[65], Shahnaz Ibrahim[66], Farida Jan[66], Jun Mine[67], Selina Banu[68], Pierangelo Veggiotti[69], Gian V. Zuccotti[69], Michel D. Ferrari[70], Arn M.J. Van Den Maagdenberg[70], Alberto Verrotti[71], Gian L. Marseglia[72], Salvatore Savasta[72], Miguel A. Soler[73], Carmela Scuderi[74], Eugenia Borgione[74], Roberto Chimenz[75], Eloisa Gitto[75], Valeria Dipasquale[75], Alessia Sallemi[75], Monica Fusco[75], Caterina Cuppari[75], Maria C. Cutrupi[75], Martino Ruggieri[76], Armando Cama[77], Valeria Capra[77], Niccolò E. Mencacci[78], Richard Boles[79], Neerja Gupta[80], Madhulika Kabra[80], Savvas Papacostas[81], Eleni Zamba-Papanicolaou[81], Efthymios Dardiotis[82], Shazia Maqbool[83], Nuzhat Rana[84], Osama Atawneh[85], Shen Y. Lim[86], Farooq Shaikh[87], George Koutsis[88], Marianthi Breza[88], Domenico A. Coviello[89], Yves A. Dauvilliers[90], Issam AlKhawaja[91], Mariam AlKhawaja[92], Fuad Al-Mutairi[93], Tanya Stojkovic[94], Veronica Ferrucci[95], Massimo Zollo[95], Fowzan S. Alkuraya[96], Maria Kinali[97], Hamed Sherifa[98], Hanene Benrhouma[99], Ilhem B.Y. Turki[99], Meriem Tazir[100], Makram Obeid[101], Sophia Bakhtadze[102], Nebal W. Saadi[103], Maha S. Zaki[104], Chahnez C. Triki[105], Fabio Benfenati[106], Stefano Gustincich[106], Majdi Kara[107], Vincenzo Belcastro[108], Nicola Specchio[109], Giuseppe Capovilla[110], Ehsan G. Karimiani[111], Ahmed M. Salih[112], Njideka U. Okubadejo[113], Oluwadamilola O. Ojo[113], Olajumoke O. Oshinaike[113], Olapeju Oguntunde[113], Kolawole Wahab[114], Abiodun H. Bello[114], Sanni Abubakar[115], Yahaya Obiabo[116], Ernest Nwazor[117], Oluchi Ekenze[118], Uduak Williams[119], Alagoma Iyagba[120], Lolade Taiwo[121], Morenikeji Komolafe[122], Konstantin Senkevich[123], Chingiz Shashkin[124], Nazira Zharkynbekova[125], Kairgali Koneyev[126], Ganieva Manizha[127], Maksud Isrofilov[127], Ulviyya Guliyeva[128], Kamran Salayev[129], Samson Khachatryan[130], Salvatore Rossi[131], Gabriella Silvestri[131], Nourelhoda Haridy[132], Luca A. Ramenghi[133], Georgia Xiromerisiou[134], Emanuele David[135], Mhammed Aguennouz[136], Liana Fidani[137], Cleanthe Spanaki[138] & Arianna Tucci[139]

[57]Children's Hospital of Rabat, University of Rabat, 6527 Rabat, Morocco. [58]Department of Paediatrics, Medical University of Sofia, Sofia 1431, Bulgaria. [59]Centre of Human Genetics, University Hospital Liege, 4000 Liege, Belgium. [60]Department of Neurology, Medical University Innsbruck, Anichstrasse 35, 6020 Innsbruck, Austria. [61]Ospedale Pediatrico "A. Cao", Department of Biomedical Sciences, University of Cagliari, 09121 Cagliari, Italy. [62]Child and Adolescent Neuropsychiatry, University of Campania "Luigi Vanvitelli", 81100 Caserta, Italy. [63]Division of Paediatric Neurology, Dayanand Medical College & Hospital, Ludhiana, Punjab 141001, India. [64]Department of Paediatric Neurology, Children's Hospital of Lahore, Lahore 381-D/2, Pakistan. [65]Department of Medical Genetics, Aga Khan University Hospital, Karachi, Karachi City, Sindh 74800, Pakistan. [66]Department of Paediatric Neurology, Aga Khan University Hospital, Karachi, Karachi City, Sindh 74800, Pakistan. [67]Department of Pediatrics, Shimane University School of Medicine, 89-1 Enya, Izumo, Shimane 6938501, Japan. [68]Institute of Child Health and Shishu Shastho Foundation Hospital, Mirpur, Dhaka 1216, Bangladesh. [69]Vittore Buzzi Children's Hospital, Via Castelvetro 32, 20154 Milan, Italy. [70]Leiden University Medical Center, Albinusdreef 2, 2333 Leiden, Netherlands. [71]Paediatric Department, San Salvatore Hospital, University of L'Aquila, L'Aquila, Italy. [72]Department of Pediatrics, University of Pavia, IRCCS Policlinico "San Matteo", 27100 Pavia, Italy. [73]Computational Modelling of Nanoscale and Biophysical systems Laboratory, Italian Institute of Technology, Genoa, Italy. [74]Laboratorio di Neuropatologia Clinica, U.O.S. Malattie, Neuromuscolari Associazione OASI Maria SS. ONLUS – IRCCS, Via Conte Ruggero 73, 94018 Troina, Italy. [75]Department of Pediatrics, University Hospital "Gaetano Martino", University of Messina, 98123 Messina, Italy. [76]Department of Clinical and Experimental Medicine, Section of Pediatrics and Child Neuropsychiatry, University of Catania, 95124 Catania, Italy. [77]Department of Neurosurgery, IRCCS Istituto Giannina Gaslini, Genoa, Italy. [78]Department of Neurology, Northwestern University Feinberg School of Medicine, Chicago, IL 60611, USA. [79]Courtagen Life Sciences, 12 Gill Street Suite 3700, Woburn, MA 01801, USA. [80]Division of Genetics, Department of Pediatrics, All India Institute of Medical Sciences (AIIMS), New Delhi, India. [81]The Cyprus Institute of Neurology and Genetics, 1683 Nicosia, Cyprus. [82]University Hospital of Larissa, Department of Neurology, Larissa 413 34, Greece. [83]Department of Developmental and Behavioral Pediatrics, Children Hospital Complex and Institute of Child Health, Lahore 381-D/2, Pakistan. [84]Department of Pediatric Neurology, Children Hospital Complex and Institute of Child Health, Multan 60000, Pakistan. [85]Hilal Pediatric Hospital Hebron, Hebron, West Bank, Hebron 90403, Palestine. [86]Department of Biomedical Science, Faculty of Medicine, University of Malaysia, Selangor 50603, Malaysia. [87]Jeffrey Cheah School of Medicine and Health Sciences, Monash University Malaysia, Selangor 47500, Malaysia. [88]Neurogenetics Unit, Neurology Department, Eginition Hospital, National and Kapodistrian University, Athens 16121, Greece. [89]Laboratorio di Genetica Umana, IRCCS Istituto Giannina Gaslini, Genoa 16147, Italy. [90]University Hospital Montpellier, Montpellier 34080, France. [91]Albashir University Hospital, Amman 11180, Jordan. [92]Prince Hamzah Hospital, Ministry of Health, Amman 11181, Jordan. [93]King Saud

University, Riyadh 11362, Saudi Arabia. [94]Institute of Myology, Hôpital La Pitié Salpêtrière, Paris 75651, France. [95]CEINGE, Biotecnologie Avanzate S. c.a.r.l., Naples 80145, Italy. [96]King Faisal Specialist Hospital and Research Center, Riyadh 43518, Saudi Arabia. [97]The Portland Hospital, 205-209 Great Portland Street, London W1W 5AH, UK. [98]Assiut University Hospital, Assiut 71515, Egypt. [99]Research Unit UR12 SP24, Department of Child and Adolescent Neurology, National Institute Mongi Ben Hmida of Neurology, Tunisi 22252, Tunisia. [100]Laboratoire de Recherche en Neurosciences, Service de Neurologie, Algeri 19785, Tunisia. [101]Department of Anatomy, Cell Biology and Physiology, American University of Beirut Medical Center, Beirut 1107 2020, Lebanon. [102]Department of Child Neurology, Tbilisi State Medical University, Tbilisi 380077, Georgia. [103]Baghdad College of Medicine, Children Welfare Teaching Hospital, Baghdad 10007, Iraq. [104]Human Genetics and Genome Research Division, National Research Centre, Cairo 12622, Egypt. [105]Child Neurology Department, Hedi Chaker hospital- Sfax Tunisia, Sfax 3000, Tunisia. [106]Istituto Italiano di Tecnologia, Genoa 16132, Italy. [107]Paediatric Neurology Unit, Department of Pediatrics, University of Tripoli, Tripoli, Libya. [108]Neurology Unit, S. Anna Hospital, Como 22042, Italy. [109]Child Neurology Unit, "Bambino Gesù" Pediatric Hospital, Rome 00165, Italy. [110]Child Neuropsychiatry Department, Epilepsy Center, C. Poma Hospital, Mantova 46100, Italy. [111]Genetics Research Centre, Molecular and Clinical Sciences Institute, St George's, University of London, Cranmer Terrace, London SW17 0RE, UK. [112]Medical University of Duhok, Duhok, Kurdistan region 1006 AJ, Iraq. [113]College of Medicine, University of Lagos (CMUL) & Lagos University Teaching Hospital, Idi Araba, Lagos State, Nigeria. [114]University of Ilorin Teaching Hospital (UITH), Ilorin, Kwara State, Nigeria. [115]Ahmadu Bello University, Zaria, Kaduna State, Nigeria. [116]Delta State University Teaching Hospital, Oghara, Delta State, Nigeria. [117]Federal Medical Centre, Owerri, Imo State, Nigeria. [118]University of Nigeria Teaching Hospital, Ituku-Ozalla, Enugu State, Nigeria. [119]University of Calabar Teaching Hospital, Calabar, Cross Rivers State, Nigeria. [120]University of Port Harcourt Teaching Hospital, Port Harcourt, Rivers State, Nigeria. [121]Babcock University, Ilishan, Remo & Federal Medical Centre, Abeokuta 4003 Ogun State, Nigeria. [122]Obafemi Awolowo University Teaching Hospital (OAUTH), Ile-Ife, Osun State, Nigeria. [123]Almazov Medical Research Centre and Pavlov First Saint Petersburg State Medical University, Saint-Petersburg 197341, Russia. [124]Kazakh National State University, Almaty 050040, Kazakhstan. [125]Shymkent Medical Academy, Almaty 122002, Kazakhstan. [126]Kazakh National State University, Almaty 050040, Kazakhstan. [127]Avicenna Tajik State Medical University, Dushanbe D61/8, Tajikistan. [128]Mediclub clinic, Baku, Azerbaijan. [129]Azerbaijan State Medical University, Baku 15080, Azerbaijan. [130]"Somnus" Neurology Clinic Sleep and Movement Disorders Center, Yerevan 0087, Armenia. [131]Department of Neurology, Università Cattolica del Sacro Cuore, Rome 00168, Italy. [132]Department of Neurology and Psychiatry, Assuit University Hospital, Assiut 71515, Egypt. [133]Neonatal Intensive Care Unit, Istituto Giannina Gaslini, 16147 Genova, Italy. [134]Department of Neurology, Medical School, University of Thessaly, Volos 38221, Greece. [135]Radiology Unit, Papardo Hospital, Viale Ferdinando Stagno d'Alcontres, Contrada Papardo, Messina 98158, Italy. [136]Unit of Neurology and Neuromuscular Diseases, Department of Clinical and Experimental Medicine, University of Messina, 98124 Messina, Italy. [137]Department of Biology, Medical School, Aristotle University, Thessaloniki 54124, Greece. [138]Department of Neurology, Medical School, University of Crete, Heraklion 74100, Greece. [139]William Harvey Research Institute, The NIHR Biomedical Research Centre at Barts, Queen Mary University London, London E1 4NS, UK

