## [Peer Review File · Nature Communications]

Reviewers' comments:

Reviewer #1 (Remarks to the Author):

Review of the manuscript

“AMPA receptor subunits are a pivotal cause of neurodevelopmental disorders”

Salpietro et al. present 22 nonrelated individuals with heterozygous de-novo GRIA2 variants and neurodevelopmental disorder (NDD). The mutations include frameshift, stop-gain, splice-site and missense variants. The phenotype that associated with mutations include intellectual disability (ID) and in most cases autism spectrum disorder (ASD). Interestingly, Rett type features were identified in some cases and some mutations associated with epileptic encephalopathy. Functional consequences of distinct mutations were investigated, but no correlation between the phenotypes within the NDD spectrum and effects of mutation were found. However, mutations were shown to affect AMPAR signaling by different means. Increased rectification was shown to be consistent with dysregulated GluA expression. The authors discuss a possible significant role of AMPAR in NDD and complex receptor-regulating mechanisms.

The study is well conducted and as a whole, well written. Methods used are sound and statistical analysis appropriate. The results are novel and provide important information about genetic mechanism underlying neurodevelopmental disorders.

Minor comment

Since no studies of *Gria2*^{-/-} transgenic mice were performed and there appear to be major differences between human and mouse AMPAR regulation, there is no need to introduce these mice in Abstract.

Reviewer #2 (Remarks to the Author):

Salpietro, Houlden and colleagues present a study on AMPA receptors GluA2. Taken together, the gene-phenotype association is valid the manuscript should be published. There were interesting functional analyses, but these were not properly discussed. In some details:

1. Abstract: The authors spend too much text on an introductory part, but do not interpret their functional results in a proper way. Eventually, they claim to have identified the “most

important” thing about AMPAR, which is not a proper sentence in this context, and anyhow not correct. I encourage the authors to moderation.

2. Background: there is no necessity for the sentences from the start until line 96. Also, the last paragraph is a repeated abstract and thus superfluous. Either one should shorten the background significantly, or one can discuss and compare the results of the other GRIAs.

3. Results

a. The first paragraph is written in a concise and good way.

b. In the second paragraph, however, some sentences generate false impressions of the data. Examples; “Progressive microcephaly was noted in several cases” while it was noted in only 3 of 22 cases, this is a minority. Also the text suggests EE and epilepsy as a major feature, while few had epilepsy and further few had mild forms such as febrile seizures or absences. Thus, I suggest to review the clinical description in paragraph 2 of the results and to be modest.

c. The third paragraph should be a part of paragraph 1 since it describes the variants, which has been started in paragraph 1.

d. The functional analyses in the 4th paragraph of the results are not clear to me. I am not an expert in patch clamp experiments, but I think this should not be a must in order to get the message. E.g.

i. Seven missense variants have been tested. Why not all and how have these seven been chosen?

ii. The authors write that also variants associated with NDD or EE are included. All variants were associated with NDD. Is there a sense of this phrase that I do not understand?

iii. Four of the seven variants showed loss of inward currents. This is not clear from the figures 4 and 5, and leaves place for misinterpretation.

iv. The authors found that the variant G609R possibly has a negative dominant effect. Any other variants with such an effect, any speculations based on in silico analyses? This is an important statement and there is no discussion if this has an influence on the phenotype and what message we may obtain out of this result.

v. Later on, the authors discuss three variants without a clear rationale why these three variants have been chosen.

e. Nothing negative to comment for the following paragraphs on the conserved sites of the variants and on the molecular modelling and prediction on structural stability. However, I have to admit that I am not an expert of these methods.

4. Discussion: The authors discuss the plausibility of the gene for the phenotype and discuss the mice phenotypes. Although it is necessary to mention this, there is indeed no doubt regarding the plausibility of the gene for this phenotype. It would be more relevant to discuss why the variants lead to such comparable phenotypes although the effects may be different or if there is any

phenotype-genotype aspects. The SYTANLAAF variants are well studied in several GRIAs, so why not comparing these? Do the variants out of SYTANLAAF lead to the same effects? Is it the same pathology? Obviously not, because there are several LoF variants and possibly other GoF (beside the 609). Mentioning Rett or CDKL5 variants is not proper and did not lead to additional knowledge. So taken together, I think that the discussion did not use the functional data of the results and that there is still place for more sophisticated thoughts. Alternatively, the manuscript can be reduced to the description of the novel gene-phenotype association while functional aspects can be included in a future work.

5. Legends of Figures:

a. Figure 1: add numbers to the photos of panel B of figure 1, or even add the variants to the photos to make it easier to follow. Also, in panel C the variants are not given, but only the wildtype. Why?

b. Figure 2 is not informative and mixes symptoms and differential diagnoses. If this is the message of the figure, it should be deleted. If further clinical and genetic aspects can be presented in a more proper pattern (e.g. if there are any correlations between severity and variant (missense vs. truncating) or if a specific pattern of symptom spectrums are noticed, a figure may be informative. Otherwise, I do not see a need for it.

c. Figure 3: as in figure 1, it makes sense to add the patient's numbers and possibly the variants to the images. Also, to make clear which images belong to the same patient at the same time. Frames that gather images would make sense (other ways to demonstrate groups of images would also be fine). In the legend, a summary about the MRI finding would make sense. I could not conclude a common picture, but a specialized person may be able to do this, or to write clearly that there is no generalized picture and the MRI of the cranium is clinically useless.

d. Figure 4 is a nice one. I would add concluding/discussing sentences to the results. Take advantage of the figures and put the most important take home message of each figure in the legend.

Reviewer #3 (Remarks to the Author):

Salpietro et al., identified a number of novel human mutations of the AMPAR GluA2 subunit that are associated with neurodevelopmental disorders. They further demonstrated that some of these mutations could change GluA2 expression or trafficking in HEK cells. Overall this is an important study and highlights the importance of the GluA2 subunit in pathogenesis of neurodevelopmental disorders.

One major concern is the relatively limited characterizations of the identified mutations. For example, the V647L mutation did not change the whole-cell current, but in the corresponding patient, severe neurological phenotypes were reported, suggesting that there are other effects of this mutation on AMPARs that this manuscript failed to characterize. I think that it is important to provide some further characterizations (glutamate affinity, channel biophysical properties or assembly with other AMPAR subunits...) on these mutations, especially those not showing significant cellular phenotypes in whole-cell current measurement in HEK cells (i.e. P528T, P528-530 del and V647L). In addition, although the authors discussed the potential effects of these mutations on channel function (line 198-199), no evidence was provided. Stargazin was included in all ephy characterizations in HEK cells, but it is possible that some of mutants might actually have a compromised interaction with stargazin, therefore confounding the interpretation of the data. Furthermore, kainate was used throughout the paper in ephy experiments. I understand that kainate is a non-desensitizing partial agonist for AMPARs (or a full agonist for the AMPAR-stargazin complex), but is it possible that some of these mutations reported here actually change kainate sensitivity of AMPARs (therefore, the reduction of whole-cell currents may not necessarily indicate a change of trafficking, surface expression or incorporation of GluA2 with GluA1, but kainate sensitivity)? Thus, some further biophysical analysis of the mutations as mentioned above will strengthen the paper. Overall this reviewer's impression is that, this is an important study, but functional characterizations are a bit rudimentary.

RE: NCOMMS-18-23048 - AMPA receptor GluA2 subunit defects are a pivotal cause of neurodevelopmental disorders

Reviewer #1 (Remarks to the Author):

'The study is well conducted and as a whole, well written. Methods used are sound and statistical analysis appropriate. The results are novel and provide important information about genetic mechanism underlying neurodevelopmental disorders'.

Minor comment

Since no studies of Gria2^{-/-} transgenic mice were performed and there appear to be major differences between human and mouse AMPAR regulation, there is no need to introduce these mice in Abstract.

Thank you for the positive comments. We have removed the mouse reference from the abstract.

Reviewer #2 (Remarks to the Author):

'Taken together, the gene-phenotype association is valid the manuscript should be published. There were interesting functional analyses, but these were not properly discussed'. In some details:

1. Abstract: The authors spend too much text on an introductory part, but do not interpret their functional results in a proper way. Eventually, they claim to have identified the "most important" thing about AMPAR, which is not a proper sentence in this contest, and anyhow not correct. I encourage the authors to moderation.
2. Background: there is no necessity for the sentences from the start until line 96. Also, the last paragraph is a repeated abstract and thus superfluous. Either one should shorten the background significantly, or one can discuss and compare the results of the other GRIAs.
3. Results
 - a. The first paragraph is written in a concise and good way.
 - b. In the second paragraph, however, some sentences generate false impressions of the data. Examples; "Progressive microcephaly was noted in several cases" while it was noted in only 3 of 22 cases, this is a minority. Also the text suggests EE and epilepsy as a major feature, while few had epilepsy and further few had mild forms such as febrile seizures or absences. Thus, I suggest to review the clinical description in paragraph 2 of the results and to be modest.
 - c. The third paragraph should be a part of paragraph 1 since it describes the variants, which has been started in paragraph 1.
 - d. The functional analyses in the 4th paragraph of the results are not clear to me. I am not an expert in patch clamp experiments, but I think this should not be a must in order to get the message. E.g.
 - i. Seven missense variants have been tested. Why not all and how have these seven been chosen?

Thank you. We have tightened up the abstract and background, and revised the clinical descriptions of the patients. We have also extended our analysis to include all of the missense mutants found in the N-terminal domain, and linker regions or pore, a total of 11 mutations. Mutations in or near the flip splice region were excluded owing to the technical challenge of recording rapidly desensitizing currents, which we felt was not feasible for initial screening of mutants.

- ii. The authors write that also variants associated with NDD or EE are included. All variants were associated with NDD. Is there a sense of this phrase that I do not understand?

This has been clarified and re-written.

- iii. Four of the seven variants showed loss of inward currents. This is not clear from the figures 4 and 5, and leaves place for misinterpretation.

We agree that the figure arrangement made comparison difficult. We have updated and extended the electrophysiology data. These results are now displayed with all mutants together, and significance on the plots. We have also used a colour scheme to help the reader navigate through the different experiments.

- iv. The authors found that the variant G609R possibly has a negative dominant effect. Any other

variants with such an effect, any speculations based on in silico analyses? This is an important statement and there is no discussion if this has an influence on the phenotype and what message we may obtain out of this result.

Now that we have extended the analysis, we see that several mutants co-expressed with GluA1 produce non-functional channels. D302G, G609R and F644L are in different channel domains. G609R and F644L are both associated with Rett-like stereotypy, but this is also present in the individuals with 528-530PQK deletion and Q607E so we cannot conclude that there is a clear correlation between the phenotype and the in vitro effect of the different mutations.

v. Later on, the authors discuss three variants without a clear rationale why these three variants have been chosen.

We agree that this was poorly justified. Our experimental analysis now includes all the coding mutations except those in or near the flip/flop alternatively spliced region.

e. Nothing negative to comment for the following paragraphs on the conserved sites of the variants and on the molecular modelling and prediction on structural stability. However, I have to admit that I am not an expert of these methods.

4. Discussion: The authors discuss the plausibility of the gene for the phenotype and discuss the mice phenotypes. Although it is necessary to mention this, there is indeed no doubt regarding the plausibility of the gene for this phenotype. It would be more relevant to discuss why the variants lead to such comparable phenotypes although the effects may be different or if there is any phenotype-genotype aspects. The SYTANLAAF variants are well studied in several GRIAs, so why not comparing these? Do the variants out of SYTANLAAF lead to the same effects? Is it the same pathology? Obviously not, because there are several LoF variants and possibly other GoF (beside the 609).

We added a Lurcher homologue to our homomer screen (A643T, supplementary figure 2). We have also added a thorough discussion of other known mutations in this region. Now that we have analysed a larger group of mutants, it has become clear that the effect on the receptor cannot be predicted from the location of the mutation. For example, the 3 dominant-negative mutants are in quite distinct parts of the channel: the N-terminal domain (D302G), near the selectivity filter (G609R) and at the gate (F644L). Similarly, when we compare the 5 mutants in the SYTANLAAF region, we find loss of homomeric current in only 3/5, and clear loss of GluA2 expression only in one case (A639S). The summary plot Figure 6 shows that most mutants form a large cluster with rectification index slightly reduced from WT, and no pattern in the degree to which current amplitude is reduced.

Mentioning Rett or CDKL5 variants is not proper and did not lead to additional knowledge. So taken together, I think that the discussion did not use the functional data of the results and that there is still place for more sophisticated thoughts. Alternatively, the manuscript can be reduced to the description of the novel gene-phenotype association while functional aspects can be included in a future work.

We reduced and moderated the discussion on CDKL5 and Rett. We added a more extensive discussion on the gene-phenotype association, also including some correlations based also on the newly identified mutations. We now present a far more complete physiological dataset, and the discussion now addresses the consequences of mutations in each of the domains.

5. Legends of Figures:

- a. Figure 1: add numbers to the photos of panel B of figure 1, or even add the variants to the photos to make it easier to follow. Also, in panel C the variants are not given, but only the wildtype. Why?
- b. Figure 2 is not informative and mixes symptoms and differential diagnoses. If this is the message of the figure, it should be deleted. If further clinical and genetic aspects can be presented in a more proper pattern (e.g. if there are any correlations between severity and variant (missense vs. truncating) or if a specific pattern of symptom spectrums are noticed, a figure may be informative. Otherwise, I do not see a need for it.
- c. Figure 3: as in figure 1, it makes sense to add the patient's numbers and possibly the variants to the images. Also, to make clear which images belong to the same patient at the same time. Frames that gather images would make sense (other ways to demonstrate groups of images would also be fine). In the legend, a summary about the MRI finding would make sense. I could not conclude a common picture, but a specialized person may be able to do this, or to write clearly that there is no generalized picture and the MRI of the cranium is clinically useless.

d. Figure 4 is a nice one. I would add concluding/discussing sentences to the results. Take advantage of the figures and put the most important take home message of each figure in the legend.

We have now re-labelled the figures, and highlighted the key findings.

Reviewer #3 (Remarks to the Author):

'Overall this is an important study and highlights the importance of the GluA2 subunit in pathogenesis of neurodevelopmental disorders'.

One major concern is the relatively limited characterizations of the identified mutations. For example, the V647L mutation did not change the whole-cell current, but in the corresponding patient, severe neurological phenotypes were reported, suggesting that there are other effects of this mutation on AMPARs that this manuscript failed to characterize. I think that it is important to provide some further characterizations (glutamate affinity, channel biophysical properties or assembly with other AMPAR subunits...) on these mutations, especially those not showing significant cellular phenotypes in whole-cell current measurement in HEK cells (i.e. P528T, P528-530 del and V647L). In addition, although the authors discussed the potential effects of these mutations on channel function (line 198-199), no evidence was provided.

We have provided extensive additional characterisation of these mutants in the form of additional Western blots and electrophysiology. We agree that co-assembly with other subunits has better physiological relevance than simply expressing homomeric GluA2 (as most of our original experiments were). To this end, we have now examined co-expression with GluA1 for 11 different GRIA2 mutations in 4 different protein domains, including data on current amplitude, holding current, rectification, expression and effect on GluA1 expression. Additionally, we added new mutations to our GluA2 homomer current studies, which completes the comparison and demonstrates that some mutations affect homomers and co-expressed channels differently.

This extended dataset has made it clear that the mutations affect the channel via more than one pathway. For example, the reduced current in A639S-containing channels is accompanied by reduced expression. In contrast, the nearby mutation F644N suppressed currents while being robustly expressed. D302G had an effect that might be caused by poor expression or reduced gating, but its location far from the pore suggests a totally different mechanism at the molecular level.

To understand the link between the channel physiology and the disease, each mutation should ideally be considered individually, and profiled with an in-depth set of experiments possibly including but not limited to single channel or macropatch kinetics, interactions with stargazin or other accessory subunits, pharmacology and ion flux ratios. With this in mind, we performed some pilot experiments using overexpression in cultured rat cortical neurons. However, using either lipofectamine or magnetofection protocols, and using either GFP or pHluorin to identify transfected neurons, the success rate for obtaining stable recordings was low. Taking into account the cell to cell variability in EPSC amplitude and kinetics, a power calculation led to the conclusion that this approach would not be feasible in the time allowed for a revision.

**Figure 2.
Example**

cells for HEK cells expressing GRIA2 WT (grey) or V647L (pink). KA (1mM), Glutamate (3mM) and glutamate + cyclothiazide (10µM) were tested on up to 3 cells per mutant.

Figure 1 Primary cultured cortical neuron transfected with GFP and WT GluA2. The few expressing neurons did not allow for stable patch clamp recordings.

We also performed pilot experiments plating HEK cells co-transfected with PSD9-flag5, neuroligin 1AB, stargazin, GluA1-SEP-WT and GluA2-WT, on top of mature primary neuronal cultures. This approach can allow neurons to make artificial synaptic contacts onto the HEK cells (Biederer et al., 2002; Dixon et al., 2015). This approach too yielded too low a success rate for a systematic comparison across the different mutants.

Stargazin was included in all ephy characterizations in HEK cells, but it is possible that some of mutants might actually have a compromised interaction with stargazin, therefore confounding the interpretation of the data.

This is an important consideration, especially for the mutant P528T for which we are unable to show any physiological consequence. We wondered if the presence of stargazin might be masking a loss of function. However, when we expressed WT channels (both homomers and heteromers) without stargazin, we observed no current in HEK cells ($n=3$ transfections). Studies without stargazin in the literature typically use oocytes rather than HEK cells, which is arguably a less physiological background.

Furthermore, kainate was used throughout the paper in ephy experiments. I understand that kainate is a non-desensitizing partial agonist for AMPARs (or a full agonist for the AMPAR-stargazin complex), but is it possible that some of these mutations reported here actually change kainate sensitivity of AMPARs (therefore, the reduction of whole-cell currents may not necessarily indicate a change of trafficking, surface expression or incorporation of GluA2 with GluA1, but kainate sensitivity)? Thus, some further biophysical analysis of the mutations as mentioned above will strengthen the paper. Overall this reviewer's impression is that, this is an important study, but functional characterizations are a bit rudimentary.

We have performed a limited set of experiments comparing kainate-evoked responses to currents evoked by glutamate application, and also by glutamate together with the desensitization inhibitor cyclothiazide (see Reviewer Figure 1). We did not observe a marked deviation from the data obtained with KA alone, and the throughput was greatly reduced because of the need to wait more than 90 s between drugs for complete washout. Again, an informal power calculation based on our pilot data as well as the literature (Watson et al., 2017) argued that detecting a robust difference in the effects of glutamate would not be feasible in a relatively short time.

Clearly, there is a very broad range of experiments that could shed further light on the consequences of the mutations. However, we concluded that our efforts would be most fruitfully applied to providing a systematic and uniform analysis of a wider group of mutants, documenting current amplitudes with and without GluA1, and characterizing rectification. We have shown convincingly for all but one of the mutations that they confer a loss of function, and that different mutations do so through different mechanisms. We have not, hitherto, shown a clear correlation between the extent or mechanism of loss of function with disease phenotype. Of course, it should be noted that the cases were all *de novo*, and so the phenotypic diversity could reflect the effect of modifying genes. Ultimately, the complexity and variability we have identified across this group of mutations will provide a useful platform for the broader research community to develop more detailed understanding of AMPA receptor physiology in health and disease.

Additional references:

- Biederer T, Sara Y, Mozhayeva M, Atasoy D, Liu XR, Kavalali ET, Sudhof TC (2002) SynCAM, a synaptic adhesion molecule that drives synapse assembly. *Science* 297:1525-1531.
- Dixon CL, Zhang Y, Lynch JW (2015) Generation of Functional Inhibitory Synapses Incorporating Defined Combinations of GABA(A) or Glycine Receptor Subunits. *Front Mol Neurosci* 8:80.
- Watson JF, Ho H, Greger IH (2017) Synaptic transmission and plasticity require AMPA receptor anchoring via its N-terminal domain. *Elife* 6.

REVIEWERS' COMMENTS:

Reviewer #2 (Remarks to the Author):

Thank you for addressing almost all aspects. I find the manuscript proper for publication.

Best,

Rami Abou Jamra

Reviewer #3 (Remarks to the Author):

the authors have made efforts to address my concerns. They also stated a few technical difficulties to perform some of the experiments I suggested. I think that the authors should at least add a few sentences in the Discussion to discuss alternative explanations due to the experiments that you could not perform. (for example, potential disruption/impairment of the interaction between AMPAR and Stargazin, or altering Kainate sensitivity (ratio of Kainate current to glutamate current)).

Reviewer #3:

The authors have made efforts to address my concerns. They also stated a few technical difficulties to perform some of the experiments I suggested. I think that the authors should at least add a few sentences in the Discussion to discuss alternative explanations due to the experiments that you could not perform. (for example, potential disruption/impairment of the interaction between AMPAR and Stargazin or altering Kainate sensitivity (ratio of Kainate current to glutamate current)).

We now included a few sentences in the Discussion to explain the technical difficulties related to some of the requested experiments and discussed alternative explanations. Thank you.